# Gradual Domain Adaptation via Manifold-Constrained Distributionally Robust Optimization

**Amirhossein Saberi**[*,§]      Amir Najafi[†]      Amin Behjati[†,‡]      Ala Emrani[†,‡]

Yasaman Zolfi[†]      Mahdi Shadrooy[†]      Abolfazl Motahari[†]      Babak H. Khalaj[*,§]

[*] Department of Electrical Engineering,
[†] Department of Computer Engineering,
[§] Sharif Center for Information Systems and Data Science,
Sharif University of Technology, Tehran, Iran

## Abstract

The aim of this paper is to address the challenge of gradual domain adaptation within a class of manifold-constrained data distributions. In particular, we consider a sequence of $T \geq 2$ data distributions $P_1, \ldots, P_T$ undergoing a gradual shift, where each pair of consecutive measures $P_i, P_{i+1}$ are close to each other in Wasserstein distance. We have a supervised dataset of size $n$ sampled from $P_0$, while for the subsequent distributions in the sequence, only unlabeled i.i.d. samples are available. Moreover, we assume that all distributions exhibit a known favorable attribute, such as (but not limited to) having intra-class soft/hard margins. In this context, we propose a methodology rooted in Distributionally Robust Optimization (DRO) with an adaptive Wasserstein radius. We theoretically show that this method guarantees the classification error across all $P_i$s can be suitably bounded. Our bounds rely on a newly introduced *compatibility* measure, which fully characterizes the error propagation dynamics along the sequence. Specifically, for inadequately constrained distributions, the error can exponentially escalate as we progress through the gradual shifts. Conversely, for appropriately constrained distributions, the error can be demonstrated to be linear or even entirely eradicated. We have substantiated our theoretical findings through several experimental results.

## 1 Introduction

Gradual domain adaptation addresses a critical challenge in machine learning: the high cost and impracticality of continually preparing labeled datasets for training ML models. Once an initial labeled dataset is obtained through costly labor, machine learning models can use it to automatically label future unlabeled datasets—a procedure called self-training. However, as these future datasets experience gradual domain shifts from the original one, the initial dataset may become less effective, necessitating renewed human effort. Gradual domain adaptation has been proposed to mitigate this issue by learning a model on the initial dataset and then gradually adapting it to the future unlabeled data in a sequential manner. Formally, we consider a sequence of datasets modeled via empirical measures $\widehat{P}_0, \ldots, \widehat{P}_T$, where $\widehat{P}_0$ represents the initial labeled dataset and the remaining $\widehat{P}_i$ are unlabeled. Here, $T$ denotes the length of the sequence, and each $\widehat{P}_i$ is an empirical estimate of an unknown distribution $P_i$ based on $n_i$ i.i.d. samples. We assume that consecutive measures $P_i$ and

---

[*]Emails: `sah.saberi@ee.sharif.edu`, `{amir.najafi,motahari,khalaj}@sharif.edu`. (‡) Authors with equal contribution.

38th Conference on Neural Information Processing Systems (NeurIPS 2024).

$P_{i+1}$ are within a bounded Wasserstein distance from each other to make the problem theoretically approachable.

Recent research in this field has proposed various methods, each with distinct advantages and disadvantages. Theoretical advancements aim to bound the generalization error, provide robustness certificates as the model adapts to successive datasets, and, importantly, quantify the error propagation dynamics along the sequence. Naive approaches often lead to exponentially increasing errors with respect to $T$ for the model performance on the most recent dataset. For some problem families, this exponential increase is conjectured to be inevitable. However, for appropriately restricted problem sets, such as linear classifiers and distributions with hard/soft margins, novel methodologies can control error propagation [KML20]. To date, no work has fully established cases where error propagation remains fixed or increases sublinearly, and a comprehensive theoretical characterization of problems in this context is still lacking.

We aim to address these challenges with a novel approach leveraging distributionally robust optimization (DRO) for gradual domain adaptation. Our core idea is based on the limited knowledge that the unknown labeled version of distribution $P_{i+1}$, or empirically, the unlabeled measure $\widehat{P}_i$ together with its latent labels, is within a bounded proximity of $P_i$ in a distributional sense. Using DRO on $P_i$ (or its empirical version $\widehat{P}_i$) with a carefully chosen and adaptive adversarial radius, we provide theoretical guarantees on $P_{i+1}$. Furthermore, when distributions exhibit favorable properties—such as lying on a manifold of margin-based measures—we demonstrate that certified bounds on generalization across domains can be established. In order to do so, we introduce a new complexity measure, the "compatibility function," which depends on the classifier hypothesis set $\Theta$, the properties of the manifold for $P_i$s, and the Wasserstein distance between consecutive distributions $P_i$ and $P_{i+1}$. This measure effectively bounds error propagation and identifies scenarios where errors remain bounded. Our analysis also extends to non-asymptotic cases where only empirical estimates of the distributions are available, showing that error terms decrease with $[\min_i n_i]^{-1/2}$.

We apply our method theoretically to two examples: (i) a toy example involving linear classifiers and Gaussian mixture model data with two components, which has been central in previous studies on DRO and gradual domain adaptation, and (ii) a more general class of distributions (referred to as "expandable" distributions) with learnable classifiers. In the former case, we demonstrate that accounting for Gaussian structural information eliminates error propagation in the statistical sense in the asymptotic regime. Additionally, in the non-asymptotic scenario, having $n \geq dT \log T$ samples per dataset leads to the same result. Conversely, neglecting manifold information results in exponentially growing error, as anticipated. For expandable distributions and learnable classifiers, we provide theoretical bounds on sample complexity and error propagation dynamics based on newer notions of adversarial robustness. Once again, we identify a rather general scenario where DRO completely eliminates error propagation. We further validate our theoretical findings through a series of experiments.

The rest of the paper is organized as follows: Section 1.1 reviews related work. Our methodology is discussed in Section 2, where we present our main theorems. Section 3 details our results for the Gaussian setting, while a broader class of problems, termed expandable distributions and smooth classifier families, are analyzed in Section 4, including their non-asymptotic analysis in subsection 4.1. In section 5, we will be discussing our experimental results. We conclude in Section 6.

## 1.1 Previous Works

Classic unsupervised domain adaptation aims to align feature distributions between a labeled source domain and an unlabeled target domain. Generating intermediate domains can facilitate smoother adaptation, transforming the process into gradual domain adaptation. However, these intermediate domains are often unavailable. Sagawa et al. [SH22] address this by using normalizing flows to learn transformations from the target domain to a Gaussian mixture distribution through the source domain. Zhuang et al. [ZZW23] propose Gradient Flow (GGF) to generate intermediate domains, leveraging the Wasserstein gradient flow to transition from the source to the target domain, minimizing a composite energy function.

Kumar et al. [KML20] propose a gradual self-training algorithm, adapting the initial classifier using pseudo-labels from intermediate domains. They show the importance of leveraging the gradual shift structure, regularization, and label sharpening, providing a generalization bound for target domain

error. This bound is given by $e^{\mathcal{O}(T)}\left(\epsilon_0 + \mathcal{O}\left(\sqrt{n^{-1}\log T}\right)\right)$, where $\epsilon_0$ is source domain error, and $n$ is each domain's data size. Wang et al. [WLZ22] improve this approach, achieving a significantly better generalization bound $\epsilon_0 + \tilde{\mathcal{O}}\left(T\Delta + Tn^{-1/2} + (nT)^{-1/2}\right)$, where $\Delta$ is the average distance of consecutive domain distributions, and propose an optimal strategy for constructing intermediate domain paths. He et al. [HWLZ23] suggest placing intermediate domains uniformly along the Wasserstein distance between the source and target domains to minimize generalization error. The GOAT framework, based on this insight, uses optimal transport to generate intermediate domains and applies gradual self-training. Similarly, Abnar et al. [ABG+21] introduce GIFT, which creates virtual samples from intermediate distributions by interpolating representations of examples from source and target domains. Zhang et al. [ZDJZ21] propose the AuxSelfTrain framework, generating a combination of source and target data in different proportions, gradually incorporating more target data, and employing a self-training procedure.

Unsupervised domain adaptation can be viewed as a Generalized Target Shift problem. Xiao et al. [XZLS23] introduce a discriminative energy-based method for test sample adaptation in domain generalization, modeling the joint distribution of input features and labels on source domains. Kirchmeyer et al. [KRdBG21] propose the OSTAR method, using optimal transport to align pre-trained representations without enforcing domain invariance, reweighting source samples, and training a classifier on the target domain. Generative Adversarial Networks (GANs) [GPAM+14] inspire domain adaptation methods that use a feature extractor and a classifier to generate class responses, processed by a discriminator to distinguish between source and target domains. Cui et al. [CWZ+20] introduce the Gradually Vanishing Bridge (GVB) framework to reduce domain-specific characteristics and balance adversarial training, enhancing domain-invariant representations.

## 1.2 Notations and Definition

Consider $\mathcal{X}$ as a measurable space for features and let $\mathcal{Y} = \{-1, +1\}$ represent the set of possible labels in a binary classification scenario. In this regard, $\mathcal{Z} = \mathcal{X} \times \mathcal{Y}$ encompasses the entire space of feature-label pairs. We use $\mathcal{M}(\mathcal{Z})$ to denote the set of all probability measures supported on $\mathcal{Z}$. For any $p \geq 1$, let $\|\cdot\|_p$ denote the $\ell_p$-norm. Additionally, for a probability measure $P \in \mathcal{M}(\mathcal{Z})$, the notation $P_{\mathcal{X}}$ refers to the marginal distribution of $P$ on $\mathcal{X}$. Let $g : \mathbb{R} \to \mathbb{R}$ be a given function, and consider a natural number $n \in \mathbb{N}$. We define the composition of $g$ repeated $n$ times as follows:

$$g^{\bigcirc n}(\cdot) = (g \circ g \circ \ldots \circ g)(\cdot), \quad (n \text{ times}). \tag{1}$$

In order to assess the distance between any two measures $P, Q \in \mathcal{M}(\mathcal{Z})$, we use the *Wasserstein* metric. For $P, Q \in \mathcal{M}(\mathcal{Z})$, $\lambda \geq 0$ and $p, q \geq 1$, the $\lambda$-weighted $\ell_p^q$-Wasserstein distance between $P$ and $Q$ is defined as

$$\mathcal{W}_{p,\lambda}^q(P, Q) \triangleq \inf_{\mu \in \mathcal{C}(P,Q)} \mathbb{E}\left[\left\|\boldsymbol{X} - \boldsymbol{X}'\right\|_p^q + \lambda \mathbb{1}\{y \neq y'\}\right], \tag{2}$$

where $\mathcal{C}(P, Q)$ denotes the set of all couplings $\mu \in \mathcal{Z} \times \mathcal{Z}$, ensuring that $\mu(\cdot, \mathcal{Z}) = P$ and $\mu(\mathcal{Z}, \cdot) = Q$. Also, let $\ell : \mathcal{Y} \times \mathcal{Y} \to \mathbb{R}_{\geq 0}$ be a legitimate loss function, where for most of the paper we simply consider it to be $0 - 1$ loss for simplicity in the results.

## 2 The Proposed Method: Gradual Domain Adaptation via Manifold-Constrained DRO

Let's consider the distribution set $\mathcal{G} \subseteq \mathcal{M}(\mathcal{Z})$ to denote a class, or manifold, of distributions characterized by favorable properties, such as, but not restricted to, having soft or hard margins between class-conditional measures. Throughout this paper, we presume that all measures $P_i$ belong to such a class with a known property. Failing to acknowledge this assumption could render the error propagation dynamics uncontrollable (see Theorem 3.2). Before proceeding further, let us introduce the following definitions:

**Definition 2.1** (Restricted Wasserstein Ball). Assume fixed parameters $p, q \geq 1$ and $\lambda > 0$. For $\eta \geq 0$, $\mathcal{G} \subseteq \mathcal{M}(\mathcal{X} \times \mathcal{Y})$ and $P_0 \in \mathcal{G}$, let us define

$$\mathcal{B}_\eta(P_0 | \mathcal{G}) \triangleq \left\{P \in \mathcal{G} \mid \mathcal{W}_{p,\lambda}^q(P, P_0) \leq \eta\right\} \tag{3}$$

as a $\mathcal{G}$-restricted Wasserstein ball of radius $\eta$.

**Algorithm 1:** DRO-based Domain Adaptation (DRODA)

---

**Params :** $\Theta, \mathcal{G}, p, q, \lambda$, and $\eta$

**Input  :** $P_0, \{P_{i_\mathcal{X}}\}_{1:T}$

**Initialize:**

$\varepsilon_0 \longleftarrow \eta, \quad \widehat{P}_0 \longleftarrow P_0$

$\Delta_0^*, \theta_0^* \longleftarrow \left\{ \min_{\theta \in \Theta}, \operatorname*{argmin}_{\theta \in \Theta} \right\} \sup_{P \in \mathcal{B}_{\varepsilon_0}(\widehat{P}_0 | \mathcal{G})} \mathbb{E}_P \left[ \ell \left( y, h_\theta \left( \boldsymbol{X} \right) \right) \right].$

**for** $i = 1, \dots, T-1$ **do**

$\widehat{P}_i \longleftarrow P_{i_\mathcal{X}} \left( \boldsymbol{X} \right) \mathbb{1} \left( y = h_{\theta_{i-1}^*} \left( \boldsymbol{X} \right) \right), \quad \forall \left( \boldsymbol{X}, y \right) \in \mathcal{Z}$

$\varepsilon_i \longleftarrow \lambda \Delta_{i-1}^* + \eta$

$\Delta_i^*, \theta_i^* \longleftarrow \left\{ \min_{\theta \in \Theta}, \operatorname*{argmin}_{\theta \in \Theta} \right\} \sup_{P \in \mathcal{B}_{\varepsilon_i}(\widehat{P}_i | \mathcal{G})} \mathbb{E}_P \left[ \ell \left( y, h_\theta \left( \boldsymbol{X} \right) \right) \right]$

**Result:** $\theta^* \longleftarrow \theta_{T-1}^*$

---

Building upon the above definition, we introduce our method formally outlined in Algorithm 1. The essence of our approach lies in conducting DRO on a pseudo-labeled version of $P_i$ (or its empirical estimate $\widehat{P}_i$), followed by leveraging the model to assign pseudo-labels to the subsequent unlabeled distribution. However, two crucial considerations emerge: i) the adaptive adjustment of the Wasserstein radius (also known as the adversarial power of DRO) based on the robust loss incurred in the preceding stage, and ii) post pseudo-labeling, distributions are implicitly constrained to the manifold $\mathcal{G}$. This latter aspect serves as the primary mechanism for controlling error propagation within appropriately restricted scenarios.

Before delving into the theoretical guarantees, let us introduce our new complexity measure, which quantifies the relationship between a family of binary classifiers $\mathcal{H} = \{h_\theta | \theta \in \Theta\}$ and the distribution family $\mathcal{G}$. The compatibility function, essentially a bound on the manifold-constrained adversarial loss of $\mathcal{H}$ on $\mathcal{G}$, plays a pivotal role in error propagation, as elucidated in Theorem 2.3.

**Definition 2.2** (Compatibility between $\mathcal{G}$ and $\mathcal{H}$). Consider the classifier set $\mathcal{H} \triangleq \{h_\theta | \theta \in \Theta\}$, distribution manifold $\mathcal{G} \subseteq \mathcal{M}(\mathcal{Z})$, and Wasserstein metric $\mathcal{W}_{p,\lambda}^q(\cdot, \cdot)$ for $p, q \geq 1$ and $\lambda \geq 0$. We say $\mathcal{H}$ and $\mathcal{G}$ are *compatible* according to a function $g_\lambda(\cdot) : \mathbb{R}_{\geq 0} \to \mathbb{R}_{\geq 0}$, if for $\eta > 0$ and $\forall P_0 \in \mathcal{G}$ the following bound holds:

$$g_\lambda(\eta) \geq \inf_{\theta \in \Theta} \sup_{P \in \mathcal{B}_\eta(P_0 | \mathcal{G})} \mathbb{E}_p \left[ \ell \left( y, h_\theta \left( \boldsymbol{X} \right) \right) \right]. \tag{4}$$

As can be seen, $g_\lambda(0)$ represents an upper bound on the minimum achievable *non-robust* error rate across all measures within $\mathcal{G}$. Mathematically, this is expressed as:

$$g_\lambda(0) \geq \sup_{P \in \mathcal{G}} \inf_{\theta \in \Theta} \mathbb{E}_P \left[ \ell \left( y, h_\theta \left( \boldsymbol{X} \right) \right) \right]. \tag{5}$$

We declare that $\mathcal{H}$ and $\mathcal{G}$ are *perfectly compatible* if the lower bound on the r.h.s. of (5), and consequently $g_\lambda(0)$, is zero. This means for any $P \in \mathcal{G}$, at least a classifier in $\mathcal{H}$ can perform a perfect non-robust classification.

We believe the concept of "compatibility" as defined above is natural and can uniquely characterize the applicability of GDA to a problem set. For example, assume all measures in $\mathcal{G}$ exhibit some level of "cluster assumption" or have hard margins, and that $\mathcal{H}$ is rich enough to robustly classify all $P \in \mathcal{G}$ (with some margin). Then, there exists $\delta_0 > 0$ such that $g_\lambda(\delta) = 0$ for all $\delta \leq \delta_0$. We will soon see that such a property can perfectly eliminate error propagation, as long as consecutive unlabeled measures are chosen close enough as a function of $\delta_0$. More generally, the following theorem provides a general bound on the propagation of generalization error as a function of the compatibility measure in the asymptotic case where $\min_i n_i \to \infty$.

**Theorem 2.3.** *For $\lambda > 0$ and $p, q \geq 1$, assume classifier set $\mathcal{H} \triangleq \{h_\theta | \theta \in \Theta\}$ and distribution family $\mathcal{G} \subseteq \mathcal{M}(\mathcal{Z})$ are compatible according to the Wasserstein metric $\mathcal{W}_{p,\lambda}^q(\cdot, \cdot)$ and a positive*

*function $g_\lambda(\cdot)$. Additionally, for $T \geq 1$ assume a finite sequence of distributions $P_0, P_1, \ldots, P_T$ in $\mathcal{G}$, where $\mathcal{W}_{p,\lambda}^q(P_i, P_{i+1}) \leq \eta$ for $i = 0, \ldots, T-1$ and a given $\eta \geq 0$. The initial measure $P_0$ is assumed to be known, however for $i \geq 1$, we only have access to the marginals $P_{i_\mathcal{X}}$. Then, Algorithm 1 (DRODA) with parameters $\mathcal{H}, \mathcal{G}, p, q, \lambda$, and $\eta$ outputs $\theta^* = \mathscr{A}(P_0, \{P_{i_\mathcal{X}}\}_{1:T})$ which satisfies the following bound:*

$$\mathbb{E}_{P_T}\left[\ell(y, h_{\theta^*}(\boldsymbol{X}))\right] \leq [g_\lambda(2\lambda(\cdot) + \eta)]^{\bigcirc T}\left(\inf_{\theta \in \Theta} \sup_{P \in \mathcal{B}_\eta(P_0|\mathcal{G})} \mathbb{E}_P[\ell(y, h_\theta(\boldsymbol{X}))]\right),$$

*where $\bigcirc T$ implies composition of function $u \to g_\lambda(2\lambda u + \eta)$ on the input for $T$ times. The input is the restricted robust loss on $P_0$ for a Wasserstein radius of $\eta$.*

The proof can be found in the Appendix (supplementary material). As inferred from the bound, the shape of $g_\lambda$ determines the behavior of the generalization error on the last measure. For example, if $g$ increases linearly, i.e., if the robust loss increases linearly with the adversarial radius with a coefficient greater than or equal to $1/(2\lambda)$, it implies an exponential growth in the generalization error. However, if the manifold structure on $\mathcal{G}$ causes $g$ to grow linearly with a smaller coefficient, or behave similarly to a saturating (or at least sublinear) function, error propagation can be kept bounded. In this regard, the following corollary specifies the conditions under which our algorithm provides a bounded error regardless of $T$. Proof of Corollary 2.4 can be found inside the Appendix section.

**Corollary 2.4** (Elimination of Error Propagation). *Consider the setting described in Theorem 2.3. For a given hypothesis set $\mathcal{H}$, distribution manifold $\mathcal{G}$, $0 \leq \lambda < 1$ and $p, q \geq 1$, assume the compatibility function $g_\lambda$ satisfies:*

$$g_\lambda(\eta) \leq \frac{1}{3\lambda}\eta + \alpha, \quad \forall \eta \geq 0, \tag{6}$$

*where $\alpha \geq 0$ can be any fixed value. Then, for any $T \in \mathbb{N}$ we have:*

$$\mathbb{E}_{P_T}\left[\ell(y, h_{\theta^*}(\boldsymbol{X}))\right] \leq 3\left(\alpha + \frac{1}{3\lambda}\max_{i \in [T]} \mathcal{W}_{p,\lambda}^q(P_{i-1}, P_i)\right). \tag{7}$$

*which is independent of $T$ as long as consecutive pairs remain distributionally close.*

## 3 Theoretical Guarantees on Gaussian Generative Models

In the following two sections, we investigate practical and theoretically useful cases of potentially compatible pairs $\mathcal{G}$ and $\mathcal{H}$ to achieve mathematically explicit bounds. We first focus on the well-known and celebrated example of a two-component Gaussian mixture model, which has been the focus of various previous studies [KML20, CRS+19, AUH+19]. One main reason is that our results can be easily compared with those of prior works.

Mathematically, suppose that the set of features and labels, denoted as $(\boldsymbol{X}, y) \in \mathbb{R}^d \times \{0, 1\}$, originates from a Gaussian generative model. For some $L > 0$, we have:

$$\begin{cases} P(y = \pm 1) = \frac{1}{2}, \\ \boldsymbol{X}|y \sim \mathcal{N}(y\boldsymbol{\mu}, \sigma^2 I) \end{cases} \quad \text{with} \quad \|\boldsymbol{\mu}\|_2 \geq L. \tag{8}$$

This setting implies that the class-conditional density of feature vectors consists of two Gaussians with equal covariance matrices $\sigma^2 I$ and mean vectors $\boldsymbol{\mu}$ and $-\boldsymbol{\mu}$, respectively. The $\ell_2$-norm of $\boldsymbol{\mu}$ is lower-bounded by some $L > 0$ to prevent the optimal Bayes' error from converging toward 1, thus the classification remains meaningful. Throughout this section, the distribution manifold $\mathcal{G}_g = \mathcal{G}_g(L)$ refers to this class, with the vector $\boldsymbol{\mu}$ as its only degree of freedom. In this context, our goal is to find the compatibility function $g_\lambda(\cdot)$ when linear classifiers are employed. The following theorem presents one of our main results for this purpose:

**Theorem 3.1.** *For $L > 0$ and any $\lambda \geq 0$, consider the distribution manifold $\mathcal{G}_g(L)$. Then, the compatibility function between $\mathcal{G}_g$ (w.r.t. Wasserstein metric $\mathcal{W}_{2,\lambda}^1$) and the set of linear binary classifiers as $\mathcal{H}$ satisfies this bound:*

$$g_\lambda(\eta) \leq e^{-\frac{L^2}{18\sigma^2}}, \quad \forall \eta \in [0, L/3]. \tag{9}$$

*Also, for any $T \in \mathbb{N}$ and any sequence of distributions $P_1, \ldots, P_T \in \mathcal{G}_g$ with $\mathcal{W}_{2,\lambda}^1(P_i, P_{i+1}) \leq L/3$, DRODA guarantees the following error bound on the last unlabeled measure:*

$$\mathbb{E}_{P_T}\left[\ell(y, h_{\theta^*}(\boldsymbol{X}))\right] \leq e^{-\frac{L^2}{18\sigma^2}}. \tag{10}$$

Proof can be found in Appendix A. Note that there are no error propagation, and the guaranteed error term is close to the Bayes' optimal error. In fact, it can become arbitrarily close with more sophisticated mathematics, which goes beyond the scope of this work. The condition $\eta \leq L/3$ is necessary to prevent the two Gaussians from swapping, as tracking them becomes impossible if that happens.

An important question to consider is what happens to $g_\lambda$ if one does not restrict the Wasserstein ball to the manifold $\mathcal{G}_g$. In other words, assume we set $\mathcal{G}_g$ to be the entire space of measures and not the restricted Gaussian manifold considered so far. We will show that the *manifold constraint* is a key property that provides us with the desirable result of Theorem 3.1, and losing this assumption can have catastrophic consequences.

**Theorem 3.2** (Potentials for Error Propagation). *For $L > 0$, consider the Gaussian manifold $\mathcal{G}_g(L)$ versus the set of linear classifiers in $\mathcal{X}$. Also, assume Wasserstein metric $\mathcal{W}_{2,\lambda}^1$ is being employed, for any $\lambda \geq 0$. By $g_\lambda^{\mathrm{C}}(\cdot)$, let us denote the compatibility function when manifold constraint is taken into account similar to Theorem 3.1, while $g_\lambda^{\mathrm{UC}}$ represents the unconstrained compatibility function when there are no manifold constraints, i.e., $\mathcal{G}_g = \mathcal{M}(\mathcal{Z})$. Then,*

$$g_\lambda^{\mathrm{C}}(\eta) \leq e^{-L^2/(18\sigma^2)}, \quad \eta \in [0, L/3], \quad \text{and} \quad g_\lambda^{\mathrm{UC}}(\eta) \geq \Omega\left(e^{-\frac{L^2}{2\sigma^2}} + \sqrt{\eta e^{-\frac{L^2}{2\sigma^2}}}\right), \quad \forall \eta \geq 0.$$

Proof can be found in Appendix section A. We already know the generalization error from the manifold constrained version of DRODA does not propagate after $T$ iterations. However, the error term stemming from the unconstrained version can be shown to be bounded by

$$\mathbb{E}_{P_T}\left[\ell\left(y, h_{\theta^*}(\boldsymbol{X})\right)\right] \leq \mathcal{O}\left(\left(2\lambda e^{\frac{-L^2}{2\sigma^2}}\right)^2 \eta^{(1/2^T)} + e^{\frac{-L^2}{2\sigma^2}}\right), \tag{11}$$

which shows significant potential for error propagation.

The results so far are in the statistical sense, meaning that we have assumed $\min_i n_i \to \infty$. A slight variation of our bounds still applies to the non-asymptotic case, where we can propose PAC-like generalization guarantees. The following theorem is, in fact, the non-asymptotic version of Theorem 3.1 (proof is given in Appendix A):

**Theorem 3.3** (Non-asymptotic Generalization Guarantee). *In the setting of Theorem 3.1 with some $L > 0$ and any $\lambda \geq 0$, suppose we have $n_0$ labeled samples from distribution $P_0$ and $n_i$ unlabeled samples from distribution $P_i$ for $i \in [T]$. $T$ can be unbounded, but consecutive pairs $P_i, P_{i+1}$ must have a Wasserstein distance $\mathcal{W}_{2,\lambda}^1$ bounded by $L/3$. For any $\delta \in (0, 1]$ and using algorithm DRODA, the error in the last (most recent) domain with probability at least $1 - \delta$ is bounded by:*

$$\Delta_T^* \leq 2e^{-\frac{L^2}{2\sigma^2}} + \left(\frac{d\log\frac{2T}{\delta}}{n_i}\right)^{\frac{1}{4}} \sum_{i=1}^{T}\left(\frac{4L^2}{\sigma^2}e^{-\frac{L^2}{18\sigma^2}}\right)^i. \tag{12}$$

**Corollary 3.4** (Elimination of Error Propagation in Non-asymptotic Regime). *In the setting of Theorem 3.3, assume $L \geq 11\sigma$ (e.g., each component of mean vectors $\boldsymbol{\mu}$ and $-\boldsymbol{\mu}$ are larger than $11\sigma/\sqrt{d}$). Also, assume each dataset $P_i$ for $i \geq 1$ has at least $n$ unlabeled data points, where*

$$n \geq \mathcal{O}\left(\frac{d\log T}{\varepsilon^4}\right)$$

*for some $\varepsilon > 0$. Then, the following bound holds for $\Delta_T^*$ regardless of $T \geq 2$:*

$$\Delta_T^* \leq 2e^{-\frac{L^2}{2\sigma^2}} + \frac{\varepsilon}{1 - \frac{4L^2}{\sigma^2}e^{-\frac{L^2}{18\sigma^2}}}, \tag{13}$$

*which means error propagation is perfectly eradicated.*

Corollary 3.4 can be directly proved from the result of Theorem 3.3.

# 4 Expandable Distribution Manifolds and Learnable Classifiers

The class of isotropic Gaussians, while a well-known theoretical benchmark, is still a very stringent and impractical case to study. In this section, we investigate a much more general class of distribution manifold/classifier pair families and provide both asymptotic and non-asymptotic guarantees for this regime. Before introducing our target regime, let us define some required concepts. Assume $(\mathcal{X}, \Sigma)$ is a measurable space, and let $P$ be a distribution supported over $\mathcal{X}$. For $r \geq 0$, the $r$-neighborhood of a point $\boldsymbol{X} \in \mathcal{X}$, denoted by $\mathcal{N}_r(\boldsymbol{X})$, is defined as:

$$\mathcal{N}_r(\boldsymbol{X}) = \left\{ \boldsymbol{X}' \mid \|\boldsymbol{X} - \boldsymbol{X}'\|_2 \leq r \right\}. \tag{14}$$

Similarly, the $r$-neighborhood of a Borel set $A \subseteq \mathcal{X}$ (i.e., $A \in \Sigma$) is defined as:

$$\mathcal{N}_r(A) = \left\{ \boldsymbol{X}' \mid \exists \boldsymbol{X} \in A \text{ such that } \|\boldsymbol{X} - \boldsymbol{X}'\|_2 \leq r \right\}, \tag{15}$$

we also define the $\boldsymbol{\delta}$-neighborhood of a Borel set $A \subseteq \mathcal{X}$, for $\boldsymbol{\delta} \in \mathbb{R}^d$ as:

$$\mathcal{N}_{\boldsymbol{\delta}}(A) = \left\{ \boldsymbol{X}' \mid \exists \boldsymbol{X} \in A, |\alpha| \leq 1 \text{ such that } \boldsymbol{X}' = \boldsymbol{X} + \alpha\boldsymbol{\delta} \right\}. \tag{16}$$

Following [WSCM20], we define the *expansion* property as:

**Definition 4.1** $((C_1, C_2) - \text{expansion})$. For a fixed $0 < \underline{a} \leq \bar{a} < \frac{1}{2}$ and given $C_1, C_2 \geq 0$, consider $\mathcal{A} \triangleq \{A \subseteq \mathcal{X} \mid \underline{a} \leq P(A) \leq \bar{a}\}$. Then, we say a distribution $P$ has $(C_1, C_2)$-expansion property if

$$\sup_{A \subseteq \mathcal{A}} \frac{P(\mathcal{N}_r(A))}{P(A)} \leq 1 + C_1 r \quad , \quad \inf_{A \subseteq \mathcal{A}} \frac{P(\mathcal{N}_r(A))}{P(A)} \geq 1 + C_2 r,$$

for sufficiently small $r \geq 0$.

This definition extends the $(a, c)$-expansion property defined by [WSCM20]. A $(C_1, C_2)$-expandable distribution is required to have a continuous support and avoid singularity, aligning with the majority of practical measures. Expandable distributions can be further restricted to have additional theoretical properties, such as $\epsilon$-smoothness, defined as follows:

**Definition 4.2** $(\epsilon - \text{smoothness})$. We say that a distribution $P$ supported on a feature-label space $\mathbb{R}^d \times \{\pm 1\}$ satisfies the $\epsilon$-smoothness property if for all $A \in \mathcal{A}$, there exists a constant $C$ which depends only on $P(A)$, where the class-conditional measures of $P$, i.e., $P^+(\boldsymbol{X})$ and $P^-(\boldsymbol{X})$, satisfy the following for sufficiently small $r \geq 0$:

$$\frac{1}{r}\left( \frac{P^s(\mathcal{N}_{\boldsymbol{\delta}}(A))}{P^s(A)} - 1 \right) \asymp C_A(1 \pm \epsilon), \quad \forall s \in \{\pm\}, \ \boldsymbol{\delta} \in \mathcal{X}, \ \|\boldsymbol{\delta}\|_2 \leq r. \tag{17}$$

Another necessary definition ensures that a classifier family $\mathcal{H}$ is inherently capable of achieving a low classification error on a distribution, i.e., a low bias for $\mathcal{H}$ and simultaneously a small Bayes' error for $P$.

**Definition 4.3** $(\alpha - \text{separation})$. For $\alpha \geq 0$, a distribution $P$ supported on feature-label set $\mathbb{R}^d \times \{\pm 1\}$ has the $\alpha - \text{seperation}$ property with respect to a binary classification hypothesis set $\mathcal{H}$, if

$$\inf_{h \in \mathcal{H}} P(yh(\boldsymbol{X}) \leq 0) \leq \alpha. \tag{18}$$

We can now explain our proposed setting for the expandable distribution manifold $\mathcal{G}$, which consists of expandable distributions (in both senses of $(C_1, C_2)$-expansion and $\epsilon$-smoothness, which are slight variations of each other). The core idea is to use the dual formulation of [BM19] and [GK23] for a (non-manifold constrained) Wasserstein DRO, which can be stated as follows:

$$\sup_{P \in \mathcal{B}_\eta(P_0)} \mathbb{E}_P[\ell(\theta; \boldsymbol{Z})] = \inf_{\gamma \geq 0} \left\{ \gamma\eta + \mathbb{E}_{P_0}\left[ \sup_{\boldsymbol{Z}'} \{\ell(\theta; \boldsymbol{Z}') - \gamma c(\boldsymbol{Z}, \boldsymbol{Z}')\} \right] \right\}. \tag{19}$$

To add the manifold constraint, we propose restricting the space of adversarial examples $\boldsymbol{Z}'$ to be generated from a predetermined function class $\mathcal{F}$, where for $f \in \mathcal{F}$ we have $f : \mathcal{Z} \to \tilde{\mathcal{Z}}$. Each $f$ is a fixed mapping from the feature-label space to itself. By controlling the complexity of $\mathcal{F}$, one can limit the adversarial budget of the DRO and effectively simulate the condition of optimizing within a

Wasserstein ball in addition to some kind of "manifold constraint." Mathematically, we can replace the original dual form with the following (more restricted) formulation:

$$\sup_{P \in \mathcal{B}_\eta(P_0 | \mathcal{G})} \mathbb{E}_p \left[ \ell(\theta; \boldsymbol{Z}) \right] = \inf_{\gamma \geq 0} \sup_{f \in \mathcal{F}} \left\{ \gamma \eta + \mathbb{E}_{P_0} \left[ \{ \ell(\theta; f(\boldsymbol{Z})) - \gamma c(\boldsymbol{Z}, f(\boldsymbol{Z})) \} \right] \right\}, \qquad (20)$$

There exists a (potentially intricate) mathematical relationship between the distributional manifold $\mathcal{G}$ on the left-hand side and the mapping function class $\mathcal{F}$ on the right-hand side of the above formulation. Our main contribution in this part can be informally stated as follows:

- We theoretically show that restricting the distributional manifold $\mathcal{G}$ to include only expandable distributions, as defined in Definitions 4.1 and 4.2, is *equivalent* to restricting the dual optimization formulation such that the mapping class $\mathcal{F}$ consists only of "smooth" mappings.

Note that if we do not impose any constraints on $\mathcal{F}$, and $f$ could be any function, there is no difference between the quantities in Equations (19) and (20).

**Theorem 4.4** (Transportability of $\epsilon$-Smooth Measures). *For some $\epsilon > 0$, let us consider two data distributions $P_1$ and $P_2$, both of which are $\epsilon$-smooth according to Definition 4.2. Let $f^+$ and $f^-$ represent the optimal transport (Monge) mappings between $P_1^+ \to P_2^+$ and $P_1^- \to P_2^-$, respectively. These mappings are also known as push-forward functions, which transform one measure into another. Let $J^+$ and $J^-$ represent the respective $d \times d$ Jacobian matrices of the mappings, where $d = \dim(\mathcal{X})$. Then, the eigenvalues of the Jacobian matrices satisfy the following conditions:*

$$1 - 2\epsilon \leq \mathsf{EIG}_i(J^s) \leq 1 + 2\epsilon, \quad \forall i \in [d], \ s \in \{\pm 1\}. \qquad (21)$$

Proof is given in Appendix A. Essentially, the theorem states that each pair of $\epsilon$-smooth class-conditional measures can be optimally transported into each other via highly-smooth mappings, where the Jacobian of the mapping resembles the identity matrix. At this point, we can present a theorem that provides an upper bound for the compatibility function between a distribution class $\mathcal{G}$ with expansion properties and a hypothesis set of learnable binary classifiers:

**Theorem 4.5.** *For $C_1, C_2, \alpha, \epsilon \geq 0$, consider a distribution manifold $\mathcal{G}$ where its distributions satisfy the $(C_1, C_2)$-expansion, $\alpha$-separation with respect to a hypothesis set $\mathcal{H}$, and $\epsilon$-smoothness properties as defined in Definitions 4.1 through 4.3. Hypothesis set $\mathcal{H}$ is general up to $\alpha$-separation property. For $\lambda, \eta \geq 0$ and $T \in \mathbb{N}$, consider the GDA setting of Theorem 2.3 with a distributional sequence $P_0, \ldots, P_T \in \mathcal{G}$ where the pairwise distance between consecutive measures satisfies $\mathcal{W}_{2,\lambda}^1(P_i, P_{i+1}) \leq \eta$. Moreover, make the following assumptions: i) assume all of the mass of $P_0$ falls inside a hypersphere with radius at most $R$, ii) assume $\epsilon \leq \frac{\eta}{14R}$, and iii) $\lambda > \eta$. Then the compatibility function between $\mathcal{G}$ and $\mathcal{H}$ has the following upper bound:*

$$g_\lambda(\eta) \leq \mathcal{O} \left( (1 + C_1(4R\epsilon + 2\eta)) \alpha \right). \qquad (22)$$

Proof can be found in Appendix A. Based on the bound on the compatibility function, using Theorem 2.3, it can be easily shown that a modified version of Algorithm DRODA, presented in Appendix A in Algorithm 3, guarantees a generalization error of at most $\mathcal{O}(\alpha + \eta)$ on the last (most recent) distribution $P_T$, which is irrespective of $T$, thereby entirely eliminating error propagation.

## 4.1 Non-Asymptotic Analysis of Manifold-Constrained DRO on Expandable Distributional Manifold

This section explores the non-asymptotic analysis of manifold-constrained DRO on the expandable distribution manifold. We assume empirical estimates from $P_i$'s, where each empirical measure $\widehat{P}_i$ is obtained via $n_i$ i.i.d. samples from $P_i$. For simplicity in our results, we assume $n_i = n$ for all $i \in \{0, 1, \ldots, T\}$. First, let us redefine the loss in its dual format:

$$R(\theta; P_0) = \sup_{f \in \mathcal{F}} \mathbb{E}_{P_0} \left[ \{ \ell(\theta; f(\boldsymbol{Z})) - \gamma c(\boldsymbol{Z}, f(\boldsymbol{Z})) \} \right]. \qquad (23)$$

Assuming $R(\theta; \widehat{P}_0)$ is the empirical version of $R(\theta; P_0)$, let the minimizer of $R(\theta; \widehat{P}_0)$ be denoted as $\widehat{\theta}$. For simplicity, we only consider the class of linear binary classifiers and Gaussian mixture models. However, the main distinction between the setting of Theorem 3.3 and this section lies in the

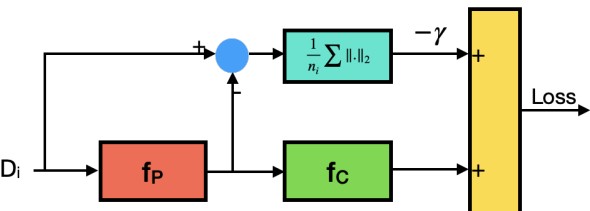

Figure 1: A schematic view of the proposed procedure for our manifold-constrained DRO. A restricted adversarial block, modeled by $f_P$, tries to perturb the source distribution at each step $i$ to prepare the algorithm for the worst possible distribution in step $i + 1$. Meanwhile, a classifier $f_C$ tries to learn a classifier based on the perturbed distribution.

fact that we assume no prior knowledge regarding the Gaussian assumption; only the expandable distribution assumption is considered. In other words, there are no implicit or explicit projection onto the manifold of Gaussian mixture models any more. At this point, we present our main theorems, which provide the generalization bound for DRODA algorithm:

**Theorem 4.6** (Generalization Bound for DRODA in the Non-asymptotic Regime). *Consider the class of linear classifiers as $\Theta$, and the zero-one loss function as $\ell$. The rest of the setting is similar to Theorem 4.5. Assume we limit $\mathcal{F}$ to the displacement functions and let $P_0$ be a Gaussian generative model with mean $\boldsymbol{\mu}$ and covariance matrix $\sigma^2 I_d$ as defined in (8), then for $\epsilon, \delta > 0$ we have the following generalization bound with probability at least $1 - \delta$:*

$$R\left(\widehat{\theta}; P_0\right) \leq \min_{\theta \in \Theta} R\left(\theta; P_0\right) + 64\sqrt{\frac{d}{n} \log\left(\frac{R}{\delta}\sqrt{\frac{n^3}{d}}\right)}, \tag{24}$$

*where $n$ is the number of i.i.d. samples from $P_0$ and $d$ is the dimension of the feature space.*

Proof can be found in Appendix A. As demonstrated, not only is error propagation eliminated, but the generalization error also decreases with increasing $n$.

The polynomial-time convergence of Wasserstein-based DRO programs have been extensively studied (see [SNVD17]). Given sufficient assumptions on the smoothness of our loss functions and transportation costs in the Wasserstein metric, the convergence rate of $\mathcal{O}\left(1/\varepsilon^2\right)$ iterations (for any $\varepsilon > 0$) in order to get to the $\varepsilon$-proximity of the optimal solution is already guaranteed.

## 5    Experimental Results

In this section we present our experimental results. It should be noted that several existing works have already experimentally validated the first part of the paper which concerns Gaussian mixture models. Hence, our contributions for those parts are mainly theoretical. In this section, we mainly focus on the second part of our contributions, i.e., Section 4.

In figure 1 we illustrate the workings of our method to generate adaptive mappings between consecutive distributions and the following projection onto the manifold, which is mathematically modeled by the function space $\mathcal{F}$ in our formulations. As depicted, at the $i$th step, we perturb the data samples $(\boldsymbol{X}_j, y_j), j \in [n_i]$ from $P_i$ using a parametric function class, denoted as $f_p$, and penalize the extent of perturbation using the following term

$$\frac{\gamma}{n_i} \sum_{j=1}^{n_i} \|f_p(\boldsymbol{X}_j) - \boldsymbol{X}_j\|_2. \tag{25}$$

These perturbed samples are then classified using a classifier. Let $L_C\left(f_P; \boldsymbol{X}_1, \cdots, \boldsymbol{X}_{n_i}\right)$ represent the cross-entropy loss of the classifier on the perturbed samples. Our objective is to solve the following program:

$$\min_{f_C \in \mathcal{C}} \max_{f_P \in \mathcal{P}} \left\{ L_C\left(f_p; \boldsymbol{X}_1, \cdots, \boldsymbol{X}_{n_i}\right) - \frac{\gamma}{n_i} \sum_{j=1}^{n_i} \|f_P(\boldsymbol{X}_j) - \boldsymbol{X}_j\|_2 \right\}, \tag{26}$$

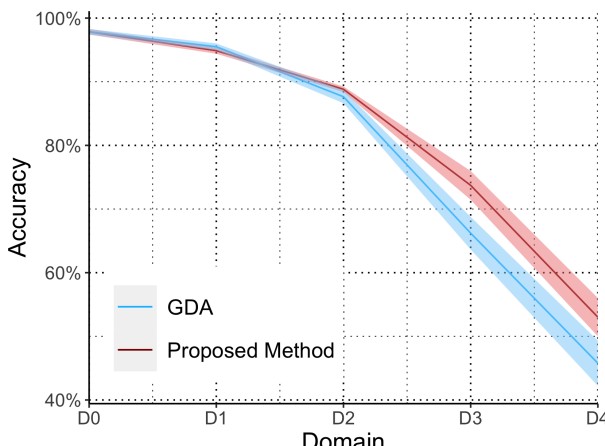

Figure 2: Comparison of the performance of our proposed method with the GDA [KML20] on rotating MNIST dataset.

which is a minimization with respect to the parametric classifier family $f_C \in \mathcal{C}$, while simultaneously maximizing it with respect to the parametric family of generator function $f_P \in \mathcal{P}$. In our experiments, we employed a two-layer CNN with a $7 \times 7$ kernel in the first layer and a $5 \times 5$ kernel in the second layer for $\mathcal{P}$. We also utilized an affine grid and grid sample function in PyTorch, following the approach introduced in [JSZ$^+$15]. For the classifier family $\mathcal{C}$, we used a three-layer CNN with max pooling and a fully connected layer, applying dropout with a rate of $0.5$ in the fully connected layer. A standard Stochastic Gradient Descent (SGD) procedure has been used for the min-max optimization procedure described in (26).

We implemented this method on the "Rotating MNIST" dataset, similar to [KML20]. In particular, we sampled 6 batches, each with a size of 4200, without replacement from the MNIST dataset, and labeled these batches as $D_0, D_1, \cdots, D_4$, which represent the datasets obtained from $P_0, P_1, \cdots, P_4$, respectively. The images in dataset $D_i$ were then rotated by $i \times 15$ degrees, with $D_0$ serving as the source dataset and $D_4$ as the target dataset. We provided the source dataset with labels and left $D_1, D_2, D_3$, and $D_4$ unlabeled for our algorithm. We then tested the accuracy of $\theta_0^*, \cdots, \theta_3^*$—the outputs of our algorithm at each step—on $D_1, D_2, D_3$, and $D_4$, respectively.

For comparison, we implemented the GDA method exactly as described in [KML20]. We compared our method to the GDA and detailed the results in Figure 2. Additionally, we reported the accuracy of $\theta_0^*$ on $D_0$ as an example of in-domain accuracy. Our results show that our method outperforms GDA by a significant margin of $8$ percent in the last domain $D_4$.

## 6 Conclusions

In conclusion, we have introduced a novel approach to gradual domain adaptation leveraging distributionally robust optimization (DRO). Our methodology provides theoretical guarantees on model adaptation across successive datasets by bounding the Wasserstein distance between consecutive distributions and ensuring that distributions lie on a manifold with favorable properties. Through theoretical analysis and experimental validation, we have demonstrated the efficacy of our approach in controlling error propagation and improving generalization across domains. A key tool for achieving this is our newly introduced complexity measure, termed the "compatibility function."

We have investigated two theoretical settings: i) a two-component Gaussian mixture model, a well-known theoretical benchmark, and ii) a more general class of distributions termed "expandable" distributions, along with general expressive (low-bias) classifier families. Theoretical analyses show that our method completely eliminates error propagation in both scenarios, and also in both asymptotic and non-asymptotic cases. These findings contribute to a better understanding of gradual domain adaptation and provide practical insights for developing robust machine learning models in real-world situations.

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

# A    Proofs for the Theorems and Corollaries

*Proof of Theorem 2.3.* The proof follows an inductive approach, relying on the steps outlined in Algorithm 1 (DRODA). To enhance clarity, we initially delve into the *step* component of the induction. Subsequently, we proceed to establish the *base* case.

**Step:**    For all $j = 1, \ldots, i$, assume $h_{\theta_{j-1}^*}$ guarantees an error rate of at most $\Delta_{j-1}^*$ on $P_j$. Then, our aim in this part of the proof is to show that:

- $h_{\theta_i^*}$ also guarantees an error rate of at most $\Delta_i^*$ on $P_{i+1}$.

- We have $\Delta_i^* \leq g_\lambda \left( 2\lambda\Delta_{i-1}^* + \eta \right)$.

Recall that for all $i = 0, 1, \ldots, T$, the marginals of $P_i$ and $\widehat{P}_i$ on $\mathcal{X}$ are the same by definition. Hence, we have
$$\mathcal{W}_p^q \left( \widehat{P}_{i_\mathcal{X}}, P_{i_\mathcal{X}} \right) = 0.$$
The conditional distribution of label $y$ given feature vector $\boldsymbol{X}$ according to $P_i$ is denoted as $P_i \left( \cdot | \boldsymbol{X} \right)$. However, again according to the definition in DRODA the conditional distribution of labels given a feature vector $\boldsymbol{X}$ for $\widehat{P}_i$ is
$$\mathbb{1} \left( \cdot = h_{\theta^* i-1} \left( \boldsymbol{X} \right) \right).$$
Let us define $Q$ as the set of all couplings between the two above-mentioned conditionals, i.e., $P_i \left( \cdot | \boldsymbol{X} \right)$ and $\mathbb{1} \left( \cdot = h_{\theta_{i-1}^*} \left( \boldsymbol{X} \right) \right)$. In this regard, we have

$$\mathcal{W}_{p,\lambda}^q \left( \widehat{P}_i, P_i \right) \leq \mathcal{W}_p^q \left( \widehat{P}_{i_\mathcal{X}}, P_{i_\mathcal{X}} \right) + \lambda \inf_{\mu \in \mathcal{Q}} \mathbb{E}_{P_{i_\mathcal{X}}} \mathbb{E}_\mu \left[ \mathbb{1} \left( y \neq y' \right) | \boldsymbol{X} \right] \leq \lambda\Delta_{i-1}^*. \tag{27}$$

Due to the triangle inequality for Wasserstein metrics, we have

$$\mathcal{W}_{p,\lambda}^q \left( \widehat{P}_i, P_{i+1} \right) \leq \mathcal{W}_{p,\lambda}^q \left( \widehat{P}_i, P_i \right) + \mathcal{W}_{p,\lambda}^q \left( P_i, P_{i+1} \right) \leq \lambda\Delta_{i-1}^* + \eta. \tag{28}$$

Consequently, having defined $\varepsilon_i \triangleq \lambda\Delta_{i-1}^* + \eta$ as in the algorithm, and noting the fact that due to the theorem's assumptions we have $P_{i+1} \in \mathcal{G}$, guarantees that $P_{i+1} \in \mathcal{B}_{\varepsilon_i} \left( \widehat{P}_i | \mathcal{G} \right)$. Therefore, we have

$$\mathbb{E}_{P_{i+1}} \left[ \ell \left( y, h_{\theta_i^*} \left( \boldsymbol{X} \right) \right) \right] \leq \Delta_i^*. \tag{29}$$

On the other hand, we have the following useful inequality for the Wasserstein balls centered on $\widehat{P}_i$ and $P_i$, respectively:

$$\mathcal{B}_{\lambda\Delta_{i-1}^*+\eta} \left( \widehat{P}_i | \mathcal{G} \right) \subseteq \mathcal{B}_{2\lambda\Delta_{i-1}^*+\eta} \left( P_i | \mathcal{G} \right), \tag{30}$$

which again directly results from triangle inequality. Therefore, we have

$$\inf_{\theta \in \Theta} \sup_{\mathcal{B}_{\lambda\Delta_{i-1}^*+\eta} \left( \widehat{P}_i | \mathcal{G} \right)} \mathbb{E}_p \left[ \ell \left( y, h_\theta \left( \boldsymbol{X} \right) \right) \right]$$

$$\leq \inf_{\theta \in \Theta} \sup_{\mathcal{B}_{2\lambda\Delta_{i-1}^*+\eta} \left( P_i | \mathcal{G} \right)} \mathbb{E}_p \left[ \ell \left( y, h_\theta \left( \boldsymbol{X} \right) \right) \right]$$

$$\leq g_\lambda \left( 2\lambda\Delta_{i-1}^* + \eta \right), \tag{31}$$

and thus $\Delta_i^* \leq g_\lambda \left( 2\lambda\Delta_{i-1}^* + \eta \right)$ which completes the *step* part of the induction.

**Base:** After the initialization step, $\theta_0^*$ represents a robust classifier that is guaranteed to have an expected error rate of $\Delta_0^* \leq g_\lambda(\eta)$ on all probability measures inside a Wasserstein ball of radius $\eta$ centered on $P_0$. This also includes $P_1$, since we have

$$\mathcal{W}_{p,\lambda}^q (P_1, P_0) \leq \eta.$$

According to DRODA, $\widehat{P}_1$ denotes a measure supported over $\mathcal{Z}$ whose marginal on $\mathcal{X}$ is the same as that of $P_1$. However, and similar to the arguments in the previous part of the proof, the conditional $P_{1_{\mathcal{Y}|\mathcal{X}}}$ has a total variation distance of at most $\lambda\Delta_0^*$ from that of $\widehat{P}_1$. Recall that the feature-conditioned distribution for labels in $\widehat{P}_1$ is a deterministic rule represented by $h_{\theta_0^*} : \mathcal{X} \to \mathcal{Y}$. In mathematical terms, the error rate on $P_1$ is guaranteed to satisfy the following upper-bound:

$$\mathbb{E}_{P_1} \left[ \ell \left( y, h_{\theta_0^*}(\boldsymbol{X}) \right) \right] \leq \Delta_0^* \leq g_\lambda(\eta). \tag{32}$$

which completes the *base* part.

**End of induction**

Combining the base with the step, one can conclude that:

$$\inf_{\theta \in \Theta} \mathbb{E}_{P_T} \left[ \ell \left( y, h_\theta(\boldsymbol{X}) \right) \right] \leq g \left( 2\lambda\Delta_{T-1}^* + \eta \right),$$

$$\Delta_{T-1}^* \leq g \left( 2\lambda\Delta_{T-2}^* + \eta \right),$$

$$\vdots$$

$$\Delta_0^* \leq \inf_{\theta \in \Theta} \sup_{P \in \mathcal{B}_\eta(P_0|\mathcal{G})} \mathbb{E}_P \left[ \ell \left( y, h_\theta(\boldsymbol{X}) \right) \right]. \tag{33}$$

Additionally, $g_\lambda(\cdot)$ is an increasing function which directly results from its definition according to Definition 2.2. This completes the whole proof. □

*Proof of Corollary 2.4.* Recall that $\alpha$ represents a value independent of $\eta$, which in fact indicates the loss of the best standard (non-robust) classifier. In general, assume we have

$$g_\lambda(\eta) \leq \beta\eta + \alpha,$$

for any fixed $\alpha, \beta \geq 0$. Then, it can be simply seen that we have

$$\begin{aligned} g_\lambda \left( 2\lambda g_\lambda(\eta) + \eta \right) &\leq g_\lambda \left( 2\lambda\beta\eta + \eta + 2\lambda\alpha \right) \\ &= g_\lambda \left( (1 + 2\lambda\beta)\eta + 2\lambda\alpha \right) \\ &\leq \beta(1 + 2\lambda\beta)\eta + (1 + 2\lambda\beta)\alpha. \end{aligned} \tag{34}$$

By induction, and assuming $2\lambda\beta < 1$, we have

$$\begin{aligned} \left[ g_\lambda \left( 2\lambda(\cdot) + \eta \right) \right]^{\bigcirc T}(\eta) &\leq \left( \sum_{i=0}^{T-1} (2\lambda\beta)^i \right) (\beta\eta + \alpha) \\ &= \frac{1 - (2\lambda\beta)^T}{1 - 2\lambda\beta} (\beta\eta + \alpha) \\ &\leq \frac{\beta\eta + \alpha}{1 - 2\lambda\beta}. \end{aligned} \tag{35}$$

Substituting with $\beta = 1/(3\lambda)$, and considering $\eta$ as a bound on the distance between consecutive distribution pairs, the result of Theorem 2.3 gives us the inequality and completes the proof. □

*Proof of Theorem 3.1.* To prove this theorem, we first need to determine the restricted Wasserstein ball for any distribution in this class. The following lemma provides a super-set for this ball:

**Lemma A.1.** *Consider a distribution $P_0 \in \mathcal{G}_g$ with parameter $\boldsymbol{\mu}_0$. Based on Definition 2.1, we have*

$$\mathcal{B}_\eta(P_0|\mathcal{G}_g) \subseteq \{ P_{\boldsymbol{\mu}} : \|\boldsymbol{\mu} - \boldsymbol{\mu}_0\|_2 \leq 2\eta \vee \|\boldsymbol{\mu} + \boldsymbol{\mu}_0\|_2 \leq 2\eta \}, \tag{36}$$

*where $P_{\boldsymbol{\mu}}$ is a Gaussian generative model with parameter $\boldsymbol{\mu}$.*

To prove this lemma we need the following lemma:

**Lemma A.2.** *Consider two arbitrary (and not necessarily Gaussian) distributions $P$ and $Q$ on $\mathcal{Z}$ with respective densities $f_1$ and $f_2$. Also, assume $P(y = 1) = Q(y = 1) = 1/2$. Let $c : \mathcal{X}^2 \to \mathbb{R}$ be a proper and lower semi-continuous transportation cost defined in the space of features, i.e., $\mathcal{X}$. For any $\lambda \geq 0$, let us define $\tilde{c} : \mathcal{Z}^2 \to \mathbb{R}$ as*

$$\tilde{c}\left(\boldsymbol{X}, y, \boldsymbol{X}', y'\right) \triangleq c\left(\boldsymbol{X}, \boldsymbol{X}'\right) + \lambda \mathbb{1}\left\{y \neq y'\right\}. \tag{37}$$

*Then, the following lower-bound holds for the Wasserstein distance between $P$ and $Q$ with respect to transportation cost $\tilde{c}$:*

$$\mathcal{W}_{\tilde{c}}\left(P, Q\right) \geq \mathcal{W}_{\tilde{c}}\left(f_1, f_2\right) \geq \frac{1}{2} \max_{i \in \{\pm 1\}} \min_{j \in \{\pm 1\}} \mathcal{W}_c\left(f_1\left(\cdot | y = i\right), f_2\left(\cdot | y' = j\right)\right). \tag{38}$$

The proofs for the above lemmas can be found in section B.

Now, we aim to find an upper bound for the compatibility function $g_\lambda\left(\cdot\right)$ between the class of Gaussian generative distributions and linear classifiers.

$$\begin{aligned}
g_\lambda\left(\eta\right) &= \max_{0 \leq i \leq T} g_\lambda^i \\
&= \max_{0 \leq i \leq T} \inf_{\theta \in \Theta} \sup_{P \in \mathcal{B}_\eta\left(P_i | \mathcal{G}\right)} \mathbb{E}_p\left[\ell\left(y, h_\theta\left(\boldsymbol{X}\right)\right)\right] \\
&\leq \max_{0 \leq i \leq T} \inf_{\theta \in \Theta} \sup_{P_{\boldsymbol{\mu}} : \|\boldsymbol{\mu} - \boldsymbol{\mu}_i\| \leq 2\eta} \mathbb{E}_{P_{\boldsymbol{\mu}}}\left[\ell\left(y, h_\theta\left(\boldsymbol{X}\right)\right)\right].
\end{aligned} \tag{39}$$

If we consider the $(0 - 1)$-loss function and the set of linear classifiers with a $d$-dimensional vector $\boldsymbol{\omega}$, where $\|\boldsymbol{\omega}\|_2 = 1$, then we have:

$$\ell\left(y, h_\theta\left(\boldsymbol{X}\right)\right) = \mathbb{1}\left(y \langle \boldsymbol{\omega}, \boldsymbol{X} \rangle \leq 0\right) \tag{40}$$

hence we can write

$$\mathbb{E}_{P_{\boldsymbol{\mu}}}\left[\ell\left(y, h_\theta\left(\boldsymbol{X}\right)\right)\right] = P_{\boldsymbol{\mu}}\left(y \langle \boldsymbol{\omega}, \boldsymbol{X} \rangle \leq 0\right) = P_{\boldsymbol{\mu}}\left(\frac{y \langle \boldsymbol{\omega}, \boldsymbol{X} \rangle}{\sigma} \leq 0\right). \tag{41}$$

We know that if $(\boldsymbol{X}, y) \sim P_{\boldsymbol{\mu}}$ then $z = \frac{y \langle \boldsymbol{\omega}, \boldsymbol{X} \rangle}{\sigma} \sim \mathcal{N}\left(\frac{\langle \boldsymbol{\omega}, \boldsymbol{\mu} \rangle}{\sigma}, 1\right)$. Therefore we can extend inequalities in (39) as follows:

$$\begin{aligned}
g_\lambda^0\left(\eta\right) &\leq \inf_{\theta \in \Theta} \sup_{P_{\boldsymbol{\mu}} : \|\boldsymbol{\mu} - \boldsymbol{\mu}_0\| \leq 2\eta} \mathbb{E}_{P_{\boldsymbol{\mu}}}\left[\ell\left(y, h_\theta\left(\boldsymbol{X}\right)\right)\right] \\
&\leq \inf_{\boldsymbol{\omega} \in \mathbb{R}^d : \|\boldsymbol{\omega}\|_2 = 1} \sup_{\boldsymbol{\mu} : \|\boldsymbol{\mu} - \boldsymbol{\mu}_0\| \leq 2\eta} \mathcal{Q}\left(\frac{\langle \boldsymbol{\omega}, \boldsymbol{\mu} \rangle}{\sigma}\right) \\
&= \inf_{\boldsymbol{\omega} \in \mathbb{R}^d : \|\boldsymbol{\omega}\|_2 = 1} \sup_{\boldsymbol{v} \in \mathbb{R}^d : \|\boldsymbol{v}\| \leq 2\eta} \mathcal{Q}\left(\frac{\langle \boldsymbol{\omega}, \boldsymbol{\mu}_0 \rangle}{\sigma} + \frac{\langle \boldsymbol{\omega}, \boldsymbol{v} \rangle}{\sigma}\right) \\
&\leq \sup_{\boldsymbol{v} \in \mathbb{R}^d : \|\boldsymbol{v}\| \leq 2\eta} \mathcal{Q}\left(\frac{\|\boldsymbol{\mu}_0\|_2}{\sigma} - \frac{\|\boldsymbol{v}\|_2}{\sigma}\right) \\
&\leq \mathcal{Q}\left(\frac{\|\boldsymbol{\mu}_0\|_2}{\sigma} - \frac{2\eta}{\sigma}\right) \\
&\leq e^{-\frac{(\|\boldsymbol{\mu}_0\|_2 - 2\eta)^2}{2\sigma^2}},
\end{aligned} \tag{42}$$

where the last inequality is true if we have:

$$\eta \leq \frac{\|\boldsymbol{\mu}_0\|_2}{2}. \tag{43}$$

Due to the above inequalities we have that $\eta \leq \frac{L}{3}$ then we have the following:

$$g_\lambda\left(\eta\right) \leq e^{-\frac{L^2}{18\sigma^2}}. \tag{44}$$

Based on the above results if we use Algorithm 1 (DRODA) when the distribution class $\mathcal{G}$, is the class of two labeled Gaussian generative model we have:

$$\mathbb{E}_{P_T}\left[\ell\left(y, h_{\theta^*}\left(\boldsymbol{X}\right)\right)\right] \leq \left[g_\lambda\left(2\lambda\left(\cdot\right)+\eta\right)\right]^{\bigcirc T}\left(\eta\right) \leq e^{-\frac{L^2}{18\sigma^2}}, \tag{45}$$

where the last inequality holds if we have:

$$2\lambda e^{-\frac{L^2}{18\sigma^2}} + \frac{L}{3} \leq \frac{L}{2} \rightarrow \lambda \leq \frac{L}{12}e^{\frac{L^2}{18\sigma^2}}.$$

We know that the error of the Bayes classifier in the target domain in the scenario of this example is equal to $e^{-\frac{\|\mu_T\|_2^2}{2\sigma^2}}$.

There is a point here that we should note. According to Lemma A.2, the last inequality in (39) is not entirely correct because it is possible for the labels of all samples to be multiplied by $-1$. To address this problem, we slightly modify our algorithm and replace the risk function as follows:

$$R\left(P, \theta\right) = \mathbb{E}_P[\ell\left(y, h_\theta(\boldsymbol{X})\right)] \rightarrow \tilde{R}\left(P, \theta\right) = \min\left\{\mathbb{E}_P[\ell\left(y, h_\theta(\boldsymbol{X})\right)], \mathbb{E}_P[\ell\left(-y, h_\theta(\boldsymbol{X})\right)]\right\}. \tag{46}$$

If we change the risk function as in the above equation, we have:

$$
\begin{aligned}
g_\lambda^i\left(\eta\right) &= \inf_{\theta\in\Theta} \sup_{P\in\mathcal{B}_\eta(P_0|\mathcal{G})} \tilde{R}\left(P, \theta\right) \\
&\leq \inf_{\theta\in\Theta} \sup_{\substack{P_{\boldsymbol{\mu}}:\|\boldsymbol{\mu}-\boldsymbol{\mu}_i\|\leq 2\eta\vee \\ \|\boldsymbol{\mu}+\boldsymbol{\mu}_i\|\leq 2\eta}} \tilde{R}\left(P_{\boldsymbol{\mu}}, \theta\right) \\
&= \inf_{\theta\in\Theta} \max_{i\in\{-1,+1\}} \sup_{P_{\boldsymbol{\mu}}:\|i\boldsymbol{\mu}-\boldsymbol{\mu}_i\|\leq 2\eta} \tilde{R}\left(P_{i\boldsymbol{\mu}}, \theta\right) \\
&\leq \inf_{\boldsymbol{\omega}\in\mathbb{R}^d:\|\boldsymbol{\omega}\|_2=1} \max_{i\in\{-1,+1\}} \sup_{P_{\boldsymbol{\mu}}:\|i\boldsymbol{\mu}-\boldsymbol{\mu}_i\|\leq 2\eta} \min\left\{\mathcal{Q}\left(\frac{\langle\boldsymbol{\omega}, i\boldsymbol{\mu}\rangle}{\sigma}\right), \mathcal{Q}\left(-\frac{\langle\boldsymbol{\omega}, i\boldsymbol{\mu}\rangle}{\sigma}\right)\right\} \\
&= \inf_{\boldsymbol{\omega}\in\mathbb{R}^d:\|\boldsymbol{\omega}\|_2=1} \sup_{P_{\boldsymbol{\mu}}:\|\boldsymbol{\mu}-\boldsymbol{\mu}_i\|\leq 2\eta} \mathcal{Q}\left(|\frac{\langle\boldsymbol{\omega}, \boldsymbol{\mu}\rangle}{\sigma}|\right) \\
&\leq e^{-\frac{(\|\boldsymbol{\mu}_i\|_2-2\eta)^2}{2\sigma^2}}. 
\end{aligned} \tag{47}
$$

The above inequality means that the result of the algorithm DRODA, with this new risk function, on the $i$th distribution has an error less than $g_\lambda^i$ either on $P_{i+1}$ or $P_{i+1}^{-1}$. Here, $P_i^{-1}$ refers to the distribution $P_i$ with its labels flipped. On the other hand, from the definition of Wasserstein distance in 2, for the class of distributions here, we have:

$$\mathcal{W}_{p,\lambda}^q\left(P_i, P_{i+1}\right) = \mathcal{W}_{p,\lambda}^q\left(P_i^{-1}, P_{i+1}^{-1}\right). \tag{48}$$

Based on the above statements the error of the output of the algorithm in the target domain can be described as follows:

$$\min\{\mathbb{E}_{P_T}\left[\ell\left(y, h_{\theta^*}\left(\boldsymbol{X}\right)\right)\right], \mathbb{E}_{P_T}\left[\ell\left(-y, h_{\theta^*}\left(\boldsymbol{X}\right)\right)\right]\} \leq e^{-\frac{L^2}{18\sigma^2}}, \tag{49}$$

This implies that if we consider the labeling by the algorithm or multiply this labeling by -1, one of these two will have an error bound as described above in the target domain. If we have one sample in the target domain, we can choose the better classifier between these two with high probability. $\square$

*Proof of Theorem 3.2.* For the constrained version, the proof is similar to the one in Theorem 3.1. Here, we present the proof for the unconstrained version of the compatibility function.

Due to [BM19] and [GK23] we know that the following holds for conntinuous $\ell$ and $c$

$$\sup_{P\in\mathcal{B}_\eta(P_0)} \mathbb{E}_p\left[\ell\left(y, h_\theta\left(\boldsymbol{X}\right)\right)\right] = \inf_{\gamma\geq 0}\left\{\gamma\eta + \mathbb{E}_{P_0}\left[\sup_{\boldsymbol{X}', y'}\left\{\ell\left(y, h_\theta\left(\boldsymbol{X}\right)\right) - \gamma\|\boldsymbol{X}-\boldsymbol{X}'\|_2 - \lambda\gamma|y-y'|\right\}\right]\right\}$$

On the other hand If we set $\lambda=\infty$ it will give a lower bound for the above quantity.

$$\sup_{P\in\mathcal{B}_\eta(P_0)} \mathbb{E}_p\left[\ell\left(y, h_\theta\left(\boldsymbol{X}\right)\right)\right] \geq \inf_{\gamma\geq 0}\left\{\gamma\eta + \mathbb{E}_{P_0}\left[\sup_{\boldsymbol{X}'}\left\{\ell\left(y, h_\theta\left(\boldsymbol{X}\right)\right) - \gamma\|\boldsymbol{X}-\boldsymbol{X}'\|_2\right\}\right]\right\} \tag{50}$$

The Problem is that the $(0-1)$-loss is not continuous. To Address this problem, we introduce a modified version of the $(0-1)$-loss function. Let us define the $(0-1)$-loss function for the class of linear classifiers as follows:

$$\ell\left(y, h_\theta\left(\boldsymbol{X}\right)\right) = \mathbb{1}\left(y h_\theta\left(\boldsymbol{X}\right) \le 0\right)$$
$$= \mathbb{1}\left(y\langle\theta, \boldsymbol{X}\rangle \le 0\right). \tag{51}$$

Now, we define the modified version of the $(0-1)$-loss function, $\ell_{\alpha,\beta}$, as follows:

$$\ell_{\alpha,\beta}^1(x) = \max\{1 - \frac{x-\beta}{\alpha}, 0\},$$
$$\ell_{\alpha,\beta}^2(x) = \min\{\ell_{\alpha,\beta}^1(x), 1\},$$
$$\ell_{\alpha,\beta}\left(y, h_\theta\left(\boldsymbol{X}\right)\right) = \ell_{\alpha,\beta}^2\left(y\langle\theta, \boldsymbol{X}\rangle\right). \tag{52}$$

From the above definitions, it can be seen that $\ell_{\alpha,-\alpha}$ is continuous and always less than or equal to the $(0-1)$-loss function. To provide a lower bound for $g_\lambda^{\mathrm{UC}}$, we consider the scenario where $\lambda = \infty$ and replace the $(0-1)$-loss function with $\ell_{\alpha,-\alpha}$ for some small positive $\alpha$.

$$g_\lambda^{\mathrm{UC}}(\eta) = \inf_{\theta\in\Theta} \sup_{P\in\mathcal{B}_\eta(P_\mu)} \mathbb{E}_p\left[\ell\left(y, h_\theta\left(\boldsymbol{X}\right)\right)\right]$$

$$\ge \inf_{\theta\in\Theta} \sup_{P\in\mathcal{B}_\eta(P_\mu)} \mathbb{E}_p\left[\ell_{\alpha,-\alpha}\left(y, h_\theta\left(\boldsymbol{X}\right)\right)\right]$$

$$= \inf_{\theta\in\Theta} \inf_{\gamma\ge0} \left\{\gamma\eta + \mathbb{E}_{P_\mu}\left[\max_{\boldsymbol{X}'} \left\{\ell_{\alpha,-\alpha}\left(y, h_\theta\left(\boldsymbol{X}\right)\right) - \gamma c\left(\boldsymbol{X}, \boldsymbol{X}'\right)\right\}\right]\right\}. \tag{53}$$

Now suppose that $c\left(\boldsymbol{X}, \boldsymbol{X}'\right) = \|\boldsymbol{X} - \boldsymbol{X}'\|_2$ and $\gamma \le \frac{1}{\alpha}$, then we can continue the above inequalities as follows:

$$g_\lambda^{\mathrm{UC}}(\eta) \ge \inf_{\theta\in\Theta} \inf_{\gamma\ge0} \left\{\gamma\eta + \mathbb{E}_{P_\mu}\left[\max_{\boldsymbol{X}'} \left\{\ell_{\alpha,-\alpha}\left(y, h_\theta\left(\boldsymbol{X}\right)\right) - \gamma c\|\boldsymbol{X} - \boldsymbol{X}'\|_2\right\}\right]\right\}$$

$$\ge \inf_{\theta\in\Theta} \inf_{\gamma\ge0} \left\{\gamma\eta + \mathbb{E}_{P_\mu}\left[\ell_{1/\gamma,-\alpha}\left(y, h_\theta\left(\boldsymbol{X}\right)\right)\right]\right\}$$

$$\ge \inf_{\gamma\ge0} \left\{\gamma\eta + \frac{e^{-\frac{(\|\mu\|_2+\alpha)^2}{2\sigma^2}}}{4\gamma\sigma} + e^{-\frac{(\|\mu\|_2+\alpha)^2}{2\sigma^2}}\right\}$$

$$\ge \Omega\left(e^{-\frac{(\|\mu\|_2+\alpha)^2}{2\sigma^2}} + \sqrt{\frac{\eta}{\sigma}}e^{-\frac{(\|\mu\|_2+\alpha)^2}{2\sigma^2}}\right). \tag{54}$$

The above inequality holds for all $\alpha \le \sqrt{\frac{e^{-\frac{|\mu|_2}{2\sigma^2}}}{4\eta\sigma}}$. Therefore, the bound is valid as we let $\alpha$ approach zero. On the other hand if in some step $i$ we have $\boldsymbol{\mu}_i = L$ then we have:

$$g_\lambda^{\mathrm{UC}}(\eta) \ge \Omega\left(e^{-\frac{L^2}{2\sigma^2}} + \sqrt{e^{-\frac{L^2}{2\sigma^2}}\eta}\right). \tag{55}$$

A very important point of Theorem 3.2 is that, constraining the Wasserstein ball could significantly improve the compatibility function $g_\lambda$. For example, if we do not constrain the Wasserstein ball and use $g_\lambda^{\mathrm{UC}}$ for the gradual domain adaptation, we can not guarantee a good upper-bound for the expected loss in the target domain, and if we use the proposed algorithm in this situation our upper-bound will be worse than the following in the target domain:

$$\mathbb{E}_{P_T}\left[\ell\left(y, h_{\theta^*}\left(\boldsymbol{X}\right)\right)\right] \le \left[g_\lambda^{\mathrm{UC}}\left(2\lambda\left(\cdot\right) + \eta\right)\right]^{\bigcirc T}\left(\inf_{\theta\in\Theta} \sup_{P\in\mathcal{B}_\eta(P_0)} \mathbb{E}_P\left[\ell\left(y, h_\theta\left(\boldsymbol{X}\right)\right)\right]\right)$$

$$\le \mathcal{O}\left(\left(2\lambda e^{\frac{L^2}{2\sigma^2}}\right)^2 \eta^{\frac{1}{2^T}} + e^{\frac{L^2}{2\sigma^2}}\right), \tag{56}$$

**Algorithm 2:** Non-asymptotic DRO-based Domain Adaptation (DRODA)

---

**Params:** $\Theta, \mathcal{G}, p, q, \lambda$, and $\eta$

**Input** : $P_0, \{P_{i_\mathcal{X}}\}_{1:T}$

**Initialize:**

$$\varepsilon_0 \longleftarrow \eta, \quad \widehat{\boldsymbol{\mu}}_0 \longleftarrow \mathbb{E}_{\widehat{P}_0}[y\boldsymbol{X}]$$

$$\Delta_0^*, \theta_0^* \longleftarrow \left\{\min_{\theta\in\Theta}, \operatorname*{argmin}_{\theta\in\Theta}\right\} \sup_{\substack{P=\mathcal{N}(y\boldsymbol{\mu},\sigma^2 I_d) \\ \|\boldsymbol{\mu}-\widehat{\boldsymbol{\mu}}_0\|\leq\varepsilon_0}} \mathbb{E}_P[\ell(y, h_\theta(\boldsymbol{X}))].$$

**for** $i = 1, \ldots, T-1$ **do**

$$\widehat{P}_i \longleftarrow \widehat{P}_{i_\mathcal{X}}(\boldsymbol{X})\mathbb{1}\left(y = h_{\theta_{i-1}^*}(\boldsymbol{X})\right), \quad \forall (\boldsymbol{X}, y) \in \mathcal{Z}$$

$$\widehat{\boldsymbol{\mu}}_i \longleftarrow \mathbb{E}_{\widehat{P}_i}[y\boldsymbol{X}]$$

$$\varepsilon_i \longleftarrow \eta + \sigma\sqrt{\frac{d\log\frac{2}{\delta}}{n_i}} + \sigma e^{\frac{-\|\mu_i\|_2^2}{2\sigma^2}}\left(1 + \Delta_{i-1}^*\right)$$

$$\Delta_i^*, \theta_i^* \longleftarrow \left\{\min_{\theta\in\Theta}, \operatorname*{argmin}_{\theta\in\Theta}\right\} \sup_{\substack{P=\mathcal{N}(y\boldsymbol{\mu},\sigma^2 I_d) \\ \|\boldsymbol{\mu}-\widehat{\boldsymbol{\mu}}_i\|\leq\varepsilon_i}} \mathbb{E}_P[\ell(y, h_\theta(\boldsymbol{X}))].$$

**Result:** $\theta^* \longleftarrow \theta_{T-1}^*$

---

where, by increasing $T$, the upper bound will be independent of $\eta$. On the other hand if we constrain the Wasserstein ball and use $g_\lambda^{\mathrm{C}}$ for the gradual domain adaptation, we can guarantee an upper-bound as good as the following for the expected loss in the target domain:

$$\mathbb{E}_{P_T}[\ell(y, h_{\theta^*}(\boldsymbol{X}))] \leq \left[g_\lambda^{\mathrm{C}}(2\lambda(\cdot) + \eta)\right]^{\bigcirc T}\left(\inf_{\theta\in\Theta}\sup_{P\in\mathcal{B}_\eta(P_0)}\mathbb{E}_P[\ell(y, h_\theta(\boldsymbol{X}))]\right)$$

$$\leq \mathcal{O}\left(e^{-\frac{(\|\mu\|-2\eta)^2}{2\sigma^2}}\right). \tag{57}$$

And the proof is complete. $\qquad\qquad\square$

*Proof of Theorem 3.3.* To prove this theorem, we first present the non-asymptotic version of Algorithm DRODA in 2. As can be seen, we have made some modifications to the algorithm in 1 to make it suitable for the non-asymptotic regime. The main idea is that, since we know the class of distributions are Gaussian generative models with different means, in each step we define a ball in which the mean of the distribution $P_i$ is contained. To do this we first bound the distance between $\widehat{\boldsymbol{\mu}}_i$ and $\boldsymbol{\mu}_i$. Where $\widehat{\boldsymbol{\mu}}_i$ is defined in the algorithm 1 and $\boldsymbol{\mu}_i$ is the mean of $i$th distribution.

$$\|\widehat{\boldsymbol{\mu}}_i - \boldsymbol{\mu}_i\|_2 = \|\widehat{\boldsymbol{\mu}}_i - \mathbb{E}_{P_{i_\mathcal{X}}}[\widehat{\boldsymbol{\mu}}_i] + \mathbb{E}_{P_{i_\mathcal{X}}}[\widehat{\boldsymbol{\mu}}_i] - \boldsymbol{\mu}_i\|_2$$

$$\leq \|\widehat{\boldsymbol{\mu}}_i - \mathbb{E}_{P_{i_\mathcal{X}}}[\widehat{\boldsymbol{\mu}}_i]\|_2 + \|\mathbb{E}_{P_{i_\mathcal{X}}}[\widehat{\boldsymbol{\mu}}_i] - \boldsymbol{\mu}_i\|_2$$

$$\leq \mathrm{I} + \mathrm{II}. \tag{58}$$

To give an upper bound for I and II in the above inequality, we should first analyze the distribution $\tilde{P}_i(\boldsymbol{X}, y) = P_{i_\mathcal{X}}(\boldsymbol{X})\mathbb{1}(y = h_{\theta_{i-1}^*}(\boldsymbol{X}))$. According to the theorem, $P_i$ is a Gaussian generative model with mean $\boldsymbol{\mu}_i$. Thus, if $(\boldsymbol{X}, y) \sim P_i$, we have $w = y\boldsymbol{X} \sim \mathcal{N}(\boldsymbol{\mu}_i, \sigma^2 I_d)$. Also, we know that $h_{\theta_{i-1}^*}$ is a linear classifier with parameter $\theta_{i-1}^*$ and error $\Delta_i^*$. We know that, $\tilde{\boldsymbol{\mu}}_i = \mathbb{E}_{\tilde{P}_i}[y\boldsymbol{X}] = \mathbb{E}_{P_{i_\mathcal{X}}}[\widehat{\boldsymbol{\mu}}_i]$.

Therefore, we have:

$$\tilde{\boldsymbol{\mu}}_i = \mathbb{E}_{P_{i\mathcal{X}}}[\text{sign}\left(\langle\theta_{i-1}^*,\boldsymbol{X}\rangle\right)\boldsymbol{X}]$$

$$= \frac{1}{2}\mathbb{E}_{\mathcal{N}(\boldsymbol{\mu}_i,\sigma^2 I_d)}[\text{sign}\left(\langle\theta_{i-1}^*,\boldsymbol{X}\rangle\right)\boldsymbol{X}] + \frac{1}{2}\mathbb{E}_{\mathcal{N}(-\boldsymbol{\mu}_i,\sigma^2 I_d)}[\text{sign}\left(\langle\theta_{i-1}^*,\boldsymbol{X}\rangle\right)\boldsymbol{X}]$$

$$= \frac{1}{2}\mathbb{E}_{\mathcal{N}(\boldsymbol{\mu}_i,\sigma^2 I_d)}[\text{sign}\left(\langle\theta_{i-1}^*,\boldsymbol{X}\rangle\right)\boldsymbol{X}] + \frac{1}{2}\mathbb{E}_{\mathcal{N}(\boldsymbol{\mu}_i,\sigma^2 I_d)}[\text{sign}\left(\langle\theta_{i-1}^*,(-\boldsymbol{X})\rangle\right)(-\boldsymbol{X})]$$

$$= \mathbb{E}_{\mathcal{N}(\boldsymbol{\mu}_i,\sigma^2 I_d)}[\text{sign}\left(\langle\theta_{i-1}^*,\boldsymbol{X}\rangle\right)\boldsymbol{X}]$$

$$= \mathbb{E}_{\mathcal{N}(0,I_d)}[\text{sign}\left(\langle\theta_{i-1}^*,(\boldsymbol{\mu}_i+\sigma\boldsymbol{u})\rangle\right)(\boldsymbol{\mu}_i+\sigma\boldsymbol{u})]$$

$$= \boldsymbol{\mu}_i\mathbb{E}_{\mathcal{N}(0,I_d)}[\text{sign}\left(\langle\theta_{i-1}^*,(\boldsymbol{\mu}_i+\sigma\boldsymbol{u})\rangle\right)] + \sigma\mathbb{E}_{\mathcal{N}(0,I_d)}[\text{sign}\left(\langle\theta_{i-1}^*,(\boldsymbol{\mu}_i+\sigma\boldsymbol{u})\rangle\right)\boldsymbol{u}], \quad (59)$$

where for the first term in the last line of the above equations we have:

$$\mathbb{E}_{\mathcal{N}(0,I_d)}[\text{sign}\left(\langle\theta_{i-1}^*,(\boldsymbol{\mu}_i+\sigma\boldsymbol{u})\rangle\right)] = \mathbb{E}_{\mathcal{N}(\boldsymbol{\mu}_i,\sigma^2 I_d)}[\text{sign}\left(\langle\theta_{i-1}^*,\boldsymbol{X}\rangle\right)]$$

$$= P_{\boldsymbol{\mu}_i}\left(\langle\theta_{i-1}^*,\boldsymbol{X}\rangle > 0\right) - P_{\boldsymbol{\mu}_i}\left(\langle\theta_{i-1}^*,\boldsymbol{X}\rangle < 0\right)$$

$$= 1 - 2P_{\boldsymbol{\mu}_i}\left(\langle\theta_{i-1}^*,\boldsymbol{X}\rangle < 0\right)$$

$$= 1 - 2P_i\left(y\langle\theta_{i-1}^*,\boldsymbol{X}\rangle < 0\right)$$

$$= 1 - 2\Delta_i, \quad (60)$$

where in the above equations $P_{\boldsymbol{\mu}_i}$ is a Gaussian probability distribution with mean $\boldsymbol{\mu}_i$ and Covariance matrix $\sigma^2 I_d$. Now we should compute the the second term in the last line of equations 59. We know that a zero mean Isotropic Gaussian random vector with identity covariance matrix is rotation invariant, therefore we can write $\boldsymbol{u} = u_\theta\widehat{\theta}_{i-1}^* + \boldsymbol{u}_\theta^\perp$, where $u_\theta \sim \mathcal{N}(0,1)$ and $\boldsymbol{u}_\theta^\perp \sim \mathcal{N}\left(0,\sigma^2 I_{d-1}\right)$, where $\widehat{\theta}_{i-1}^*$ is a vector with norm 1 in the direction of $\theta_{i-1}^*$ and $\boldsymbol{u}_\theta^\perp$ belongs to the subspace perpendicular to the $\theta_{i-1}^*$ and $u_\theta$ is independent from $\boldsymbol{u}_\theta^\perp$. Now for the second term in the last line of equations 59 we have:

$$\mathbb{E}_{\mathcal{N}(0,I_d)}[\text{sign}\left(\langle\theta_{i-1}^*,(\boldsymbol{\mu}_i+\sigma\boldsymbol{u})\rangle\right)\boldsymbol{u}] = \mathbb{E}_{\mathcal{N}(0,I_d)}[\text{sign}\left(\langle\theta_{i-1}^*,\boldsymbol{\mu}_i\rangle + \sigma u_\theta\right)\boldsymbol{u}]$$

$$= \widehat{\theta}_{i-1}^*\mathbb{E}_{\mathcal{N}(0,1)}[\text{sign}\left(\langle\theta_{i-1}^*,\boldsymbol{\mu}_i\rangle + \sigma u_\theta\right)u_\theta]$$

$$+ \mathbb{E}_{\mathcal{N}(0,1)}\mathbb{E}_{\mathcal{N}(0,I_{d-1})}[\text{sign}\left(\langle\theta_{i-1}^*,\boldsymbol{\mu}_i\rangle + \sigma u_\theta\right)\boldsymbol{u}_\theta^\perp]$$

$$= \widehat{\theta}_{i-1}^*\mathbb{E}_{\mathcal{N}(0,1)}[\text{sign}\left(\langle\theta_{i-1}^*,\boldsymbol{\mu}_i\rangle + \sigma u_\theta\right)u_\theta]$$

$$+ \mathbb{E}_{\mathcal{N}(0,1)}[\text{sign}\left(\langle\theta_{i-1}^*,\boldsymbol{\mu}_i\rangle + \sigma u_\theta\right)]\mathbb{E}_{\mathcal{N}(0,I_{d-1})}[\boldsymbol{u}_\theta^\perp]$$

$$= \widehat{\theta}_{i-1}^*\mathbb{E}_{\mathcal{N}(0,1)}[\text{sign}\left(\langle\theta_{i-1}^*,\boldsymbol{\mu}_i\rangle + \sigma u_\theta\right)u_\theta] + 0$$

$$= \widehat{\theta}_{i-1}^*\mathbb{E}_{\mathcal{N}(0,1)}\left[u_\theta\Big|u_\theta > -\frac{\langle\theta_{i-1}^*,\boldsymbol{\mu}_i\rangle}{\sigma}\right]$$

$$- \widehat{\theta}_{i-1}^*\mathbb{E}_{\mathcal{N}(0,1)}\left[u_\theta\Big|u_\theta < -\frac{\langle\theta_{i-1}^*,\boldsymbol{\mu}_i\rangle}{\sigma}\right]$$

$$= \widehat{\theta}_{i-1}^*\left(\frac{\sqrt{\frac{2}{\pi}}e^{-\frac{\langle\theta_{i-1}^*,\boldsymbol{\mu}_i\rangle^2}{2\sigma^2}}}{1-\Delta_i}\right), \quad (61)$$

where in the last line we use the fact that if $u_\theta$ has normal distribution its conditional distribution on $u_\theta > -\frac{\langle\theta_{i-1}^*,\boldsymbol{\mu}_i\rangle}{\sigma}$ and $u_\theta < -\frac{\langle\theta_{i-1}^*,\boldsymbol{\mu}_i\rangle}{\sigma}$ has truncated Gaussian distribution. Now we can continue equations in 59 as follows:

$$\tilde{\boldsymbol{\mu}}_i = \boldsymbol{\mu}_i\left(1-2\Delta_i\right) + \widehat{\theta}_{i-1}^*\left(\frac{\sqrt{\frac{2\sigma^2}{\pi}}e^{-\frac{\langle\theta_{i-1}^*,\boldsymbol{\mu}_i\rangle^2}{2\sigma^2}}}{1-\Delta_i}\right), \quad (62)$$

and for II in equation 58 we have:

$$\|\tilde{\boldsymbol{\mu}}_i - \boldsymbol{\mu}_i\|_2 = 2\Delta_i\|\boldsymbol{\mu}_i\|_2 + \left(\frac{\sqrt{\frac{2\sigma^2}{\pi}}e^{-\frac{\langle\theta_{i-1}^*,\boldsymbol{\mu}_i\rangle^2}{2\sigma^2}}}{1-\Delta_i}\right). \quad (63)$$

Now we try to compute I in equation 58. For $\widehat{\boldsymbol{\mu}}_i$ we have:

$$\widehat{\boldsymbol{\mu}}_i = \frac{1}{n_i} \sum_{j=1}^{n_i} \text{sign}\left(\langle \theta_{i-1}^*, \boldsymbol{X}_j \rangle\right) \boldsymbol{X}_j = \frac{1}{n_i} \sum_{j=1}^{n_i} \boldsymbol{Z}_j, \tag{64}$$

where $\boldsymbol{X}_j \sim P_{i\mathcal{X}}$ come from a mixture of two Gaussian distribution where their means are in $\boldsymbol{\mu}_i$ and $-\boldsymbol{\mu}_i$. On the other hand distribution of $\text{sign}\left(\langle \theta_{i-1}^*, \boldsymbol{X}_j \rangle\right) \boldsymbol{X}_j$ is not different when we have $\boldsymbol{X}_j \sim \mathcal{N}\left(\boldsymbol{\mu}_i, \sigma^2 I_d\right)$ from when $\boldsymbol{X}_j \sim \mathcal{N}\left(-\boldsymbol{\mu}_i, \sigma^2 I_d\right)$. Therefore distribution of $\boldsymbol{Z}_j = \text{sign}\left(\langle \theta_{i-1}^*, \boldsymbol{X}_j \rangle\right) \boldsymbol{X}_j$ when $\boldsymbol{X}_j$s are come from $P_{i\mathcal{X}}$ is not different from when $\boldsymbol{X}_j$s are come from $\mathcal{N}\left(\boldsymbol{\mu}_i, \sigma^2 I_d\right)$. For simplicity in the rest of the proof we will drop the subscript $j$ and use $\boldsymbol{Z}, \boldsymbol{X}$ instead of $\boldsymbol{Z}_j, \boldsymbol{X}_j$ respectively. Now for the variable $Z$ we have:

$$\mathbb{P}\left(\boldsymbol{Z} \middle| \langle \theta_{i-1}^*, \boldsymbol{X} \rangle > 0\right) = \mathbb{P}\left(\boldsymbol{X} \middle| \langle \theta_{i-1}^*, \boldsymbol{X} \rangle > 0\right)$$
$$= \mathbb{P}\left(\boldsymbol{\mu} + \sigma \boldsymbol{u} \middle| \langle \theta_{i-1}^*, \boldsymbol{\mu} \rangle + \sigma \langle \theta_{i-1}^*, \boldsymbol{u} \rangle > 0\right). \tag{65}$$

Now suppose that we rotate the space such that the $\theta_{i-1}^*$ align to the first dimension of the space. We know that the zero mean isotropic Gaussian is rotation invariant, therefore by this rotation distribution of $\boldsymbol{u}$ doesn't change. So we have:

$$\mathbb{P}\left(\boldsymbol{Z} \middle| \langle \theta_{i-1}^*, \boldsymbol{X} \rangle > 0\right) = \mathbb{P}\left(\boldsymbol{\mu}_\theta + \sigma \boldsymbol{u} \middle| \mu_{\theta 1} + \sigma u_1 > 0\right), \tag{66}$$

where $\boldsymbol{\mu}_\theta$ is the rotated version of $\boldsymbol{\mu}$, and $\mu_{\theta 1}$ and $u_1$ are the first dimension of $\boldsymbol{\mu}_\theta$ and $\boldsymbol{u}$ respectively. Due to the above equation if we name the random vector with distribution $\mathbb{P}\left(\boldsymbol{\mu}_\theta + \sigma \boldsymbol{u} \middle| \mu_{\theta 1} + \sigma u_1 > 0\right)$, $\boldsymbol{Z}_\theta$, its first dimension is a truncated random variable and its other dimensions are normal random variable and all of them are independent from each other. Therefore due to [HJ14] and [Wai19] $\|Z_\theta - \mathbb{E}\left[Z_\theta\right]\|_2^2$ is a sub exponential random variable with parameters $\left(8\sqrt{d}\sigma^2, 4\sigma^2\right)$. With the same method the random variable with distribution $\mathbb{P}\left(\boldsymbol{Z} \middle| \langle \theta_{i-1}^*, \boldsymbol{X} \rangle < 0\right)$ is sub exponential random variable with parameters $\left(8\sqrt{d}\sigma^2, 4\sigma^2\right)$. Therefore we have:

$$\mathbb{E}\left[e^{\lambda \|\boldsymbol{Z} - \mathbb{E}[\boldsymbol{Z}]\|_2^2}\right] = (1 - \Delta)\mathbb{E}\left[e^{\lambda \|\boldsymbol{Z} - \mathbb{E}[\boldsymbol{Z}]\|_2^2} \middle| \langle \theta_{i-1}^*, \boldsymbol{X} \rangle > 0\right] + \Delta \mathbb{E}\left[e^{\lambda \|\boldsymbol{Z} - \mathbb{E}[\boldsymbol{Z}]\|_2^2} \middle| \langle \theta_{i-1}^*, \boldsymbol{X} \rangle < 0\right], \tag{67}$$

and $\|\boldsymbol{Z} - \mathbb{E}\left[\boldsymbol{Z}\right]\|_2^2$ is sub exponential with parameters $\left(8\sqrt{d}\sigma^2, 4\sigma^2\right)$. So with probability more than $1 - \delta$ we have:

$$\|\widehat{\boldsymbol{\mu}}_i - \mathbb{E}_{P_{i\mathcal{X}}}\left[\widehat{\boldsymbol{\mu}}_i\right]\|_2^2 \leq \mathbb{E}\left[\|\widehat{\boldsymbol{\mu}}_i - \mathbb{E}_{P_{i\mathcal{X}}}\left[\widehat{\boldsymbol{\mu}}_i\right]\|_2^2\right] + \sigma^2 \left(\frac{d}{n_i} \log \frac{1}{\delta}\right)^{\frac{1}{2}}$$

$$\leq \frac{d\sigma^2}{n_i} + \sigma^2 \left(\frac{d}{n_i} \log \frac{1}{\delta}\right)^{\frac{1}{2}} \tag{68}$$

and therefore:

$$\|\widehat{\boldsymbol{\mu}}_i - \boldsymbol{\mu}_i\|_2 \leq \sigma\sqrt{\frac{d}{n_i}} + \sigma\left(\frac{d}{n_i} \log \frac{1}{\delta}\right)^{\frac{1}{4}} + 2\Delta_i \|\boldsymbol{\mu}_i\|_2 + \left(\frac{\sqrt{\frac{2\sigma^2}{\pi}} e^{-\frac{\langle \theta_{i-1}^*, \boldsymbol{\mu}_i \rangle^2}{2\sigma^2}}}{1 - \Delta_i}\right)$$

$$\leq \sigma\sqrt{\frac{d}{n_i}} + \sigma\left(\frac{d}{n_i} \log \frac{1}{\delta}\right)^{\frac{1}{4}} + 2\Delta_i \|\boldsymbol{\mu}_i\|_2 + \sqrt{\frac{2\sigma^2}{\pi}} e^{-\frac{L^2}{4\sigma^2}}$$

$$= \tilde{\epsilon}_i \tag{69}$$

Based on the Algorithm 2 we have:

$$\Delta_i^* = \min_{\theta \in \Theta} \sup_{\substack{P = \mathcal{N}\left(y\boldsymbol{\mu}, \sigma^2 I_d\right) \\ \|\boldsymbol{\mu} - \widehat{\boldsymbol{\mu}}_i\| \leq \varepsilon_i}} \mathbb{E}_P\left[\ell\left(y, h_\theta\left(\boldsymbol{X}\right)\right)\right]$$

$$\leq \min_{\theta \in \Theta} \sup_{\substack{P = \mathcal{N}\left(y\boldsymbol{\mu}, \sigma^2 I_d\right) \\ \|\boldsymbol{\mu} - \boldsymbol{\mu}_i\| \leq \varepsilon_i + \tilde{\epsilon}_i}} \mathbb{E}_P\left[\ell\left(y, h_\theta\left(\boldsymbol{X}\right)\right)\right]$$

$$\leq e^{-\frac{L^2}{18\sigma^2}} \left(1 + \frac{2L}{\sigma}\left(\sqrt{\frac{d}{n_i}} + \left(\frac{d}{n_i} \log \frac{1}{\delta}\right)^{\frac{1}{4}} + 2\frac{\Delta_{i-1} L}{\sigma} + e^{-\frac{L^2}{4\sigma^2}}\right)\right) \tag{70}$$

Therefore for the error in the target domain we have:

$$\Delta_T^* \le 2e^{-\frac{L^2}{2\sigma^2}} + \left(\frac{d\log\frac{2T}{\delta}}{n_i}\right)^{\frac{1}{4}} \sum_{i=1}^{T} \left(\frac{4L^2}{\sigma^2} e^{-\frac{L^2}{18\sigma^2}}\right)^i. \tag{71}$$

And the proof is complete. $\qquad\square$

*Proof of Theorem 4.4.* Since the proofs for both $f^+$ and $f^-$ are the same, we ignore the superscript and denote both functions simply by $f$. It should be noted that $f : \mathcal{X} \to \mathcal{X}$, and we have $\dim(\mathcal{X}) = d$.

for any given point $\boldsymbol{X}_0 \in \mathcal{X}$ and $i \in [d]$, let $\boldsymbol{u}_i(\boldsymbol{X}_0)$ denote the unitary direction vector of the $i$th eigenvector (without any particular order) of the Jacobian matrix of $f$ at position $\boldsymbol{X}_0$. Also, let $\lambda_i(\boldsymbol{X}_0)$ represent its corresponding eigenvalue. We drop the input argument $\boldsymbol{x}_0$ throughout the remainder of the proof, for the sake of simplicity.

For a sufficiently small $\Delta > 0$, we consider a $d$-dimensional Parallelepiped that has the following properties: i) it contains $\boldsymbol{X}_0$, ii) its edges are aligned with $\boldsymbol{u}_i(\boldsymbol{X}_0)$s for $i \in [d]$, and iii) the probability mass inside the Parallelepiped according to $P_1$ is $\Delta$. Let us call this Parallelepiped $A(\boldsymbol{X}_0)$. Again, we drop $\boldsymbol{X}_0$ for simplicity throughout the remainder of the proof.

Hence, for $j \in \{1, 2\}$, the probability mass of $A$ with respect to $P_j$ can be written as:

$$P_j(A) = P_j(\{\boldsymbol{X} : \boldsymbol{X} \in A\}) = \int_A d_j(\boldsymbol{X}) \mathrm{d}\boldsymbol{X}, \tag{72}$$

where $d_j$ represents the density function of $P_j$ with respect to Lebesgue measure. Suppose $\widehat{A}$ is the image of $A$ under the function $f$. For $P_1$ and $P_2$ satisfying the $\epsilon$-smoothness property and a vector $\boldsymbol{\delta} \in \mathcal{X}$ with $\|\boldsymbol{\delta}\|_2 \le r$ (for sufficiently small $r > 0$), we have

$$C(\Delta)(1-\epsilon)r \le \frac{\int_{\mathcal{N}_{\boldsymbol{\delta}}(A)} d_1(\boldsymbol{X})\mathrm{d}\boldsymbol{X}}{\int_A d_1(\boldsymbol{X})\mathrm{d}\boldsymbol{X}} - 1 \le C(\Delta)(1+\epsilon)r$$

$$C(\Delta)(1-\epsilon)r \le \frac{\int_{\mathcal{N}_{\boldsymbol{\delta}}(\widehat{A})} d_2(\boldsymbol{X})\mathrm{d}\boldsymbol{X}}{\int_{\widehat{A}} d_2(\boldsymbol{X})\mathrm{d}\boldsymbol{X}} - 1 \le C(\Delta)(1+\epsilon)r. \tag{73}$$

On the other hand, due to the fact that $P_2 = f_\# P_1$ the following holds:

$$\int_{\mathcal{N}_{\boldsymbol{\delta}}(\widehat{A})} d_2(\boldsymbol{X})\mathrm{d}\boldsymbol{X} = \int_{\mathcal{N}_{\boldsymbol{\delta}/\lambda_i}(A)} d_1(\boldsymbol{X})\mathrm{d}\boldsymbol{X} + \mathcal{O}(r^2), \quad \forall \boldsymbol{\delta} \in \mathcal{X}, \quad \|\boldsymbol{\delta}\|_2 \le r. \tag{74}$$

In the above inequality, the first order of $r$ appears in the volume over which the integral is calculated. This inequality directly results into the following bounds:

$$C(\Delta)(1-\epsilon)r \le \frac{\int_{\mathcal{N}_{\boldsymbol{\delta}}(\widehat{A})} d_2(\boldsymbol{X})d\boldsymbol{X}}{\int_{\widehat{A}} d_2(\boldsymbol{X})d\boldsymbol{X}} - 1$$

$$= \frac{\int_{\mathcal{N}_{\boldsymbol{\delta}/\lambda_i}(A)} d_1(\boldsymbol{X})d\boldsymbol{X}}{\int_A d_1(\boldsymbol{X})d\boldsymbol{X}} - 1 + \mathcal{O}\left(\frac{r^2}{\Delta}\right)$$

$$\le C(\Delta)(1+\epsilon)r/\lambda_i + \mathcal{O}\left(\frac{r^2}{\Delta}\right)$$

$$C(\Delta)(1-\epsilon)r/\lambda_i \le \frac{\int_{\mathcal{N}_{\boldsymbol{\delta}/\lambda_i}(A)} d_1(\boldsymbol{X})d\boldsymbol{X}}{\int_A d_1(\boldsymbol{X})d\boldsymbol{X}} - 1$$

$$= \frac{\int_{\mathcal{N}_{\boldsymbol{\delta}}(\widehat{A})} d_2(\boldsymbol{X})d\boldsymbol{X}}{\int_{\widehat{A}} d_2(\boldsymbol{X})d\boldsymbol{X}} - 1 + \mathcal{O}\left(\frac{r^2}{\Delta}\right)$$

$$\le C(\Delta)(1+\epsilon)r + \mathcal{O}\left(\frac{r^2}{\Delta}\right). \tag{75}$$

Now based on Equations (73) and (75) we have the following:

$$1 - 2\epsilon - \mathcal{O}\left(\frac{r}{\Delta}\right) \le \lambda_i \le 1 + 2\epsilon + \mathcal{O}\left(\frac{r}{\Delta}\right), \quad \forall i \in \{1, \dots, d\}. \tag{76}$$

If we set $r = \Delta \epsilon^2$ then we have:

$$1 - 2\epsilon \leq \lambda_i \leq 1 + 2\epsilon, \ \forall i \in \{1, \ldots, d\}. \tag{77}$$

And the proof is complete. $\qquad\square$

*Proof of Theorem 4.5.* Based on the result of Theorem 4.4, we know that if $P_0$ and $P_1$ both have the $\epsilon$-smoothness property and $P_1 = f_\# P_0$, then the eigenvalues of the Jacobian matrix of $f$ should not be far from 1. Therefore, if we define $\mathcal{F}$ as the class of functions with such Jacobian matrices, then we have:

$$\sup_{P \in \mathcal{B}_\eta(P_0|\mathcal{D})} \mathbb{E}_P \left[ \ell \left( y, h \left( \boldsymbol{X} \right) \right) \right] \leq \sup_{P \in \mathcal{B}_\eta(P_0|\mathcal{F})} \mathbb{E}_P \left[ \ell \left( y, h \left( \boldsymbol{X} \right) \right) \right], \tag{78}$$

where $\mathcal{B}_\eta \left( P_0 | \mathcal{F} \right)$ is defined mathematically as follows:

$$\mathcal{B}_\eta \left( P_0 | \mathcal{F} \right) \triangleq \left\{ P : P = f_\# P_0, f \in \mathcal{F}, \mathcal{W}^q_{p,\lambda} \left( P, P_0 \right) \leq \eta \right\}. \tag{79}$$

Now suppose that for a point $\boldsymbol{X}_0 \in \mathbb{R}^d$, we have $\| \boldsymbol{X}_0 - f \left( \boldsymbol{X}_0 \right) \|_2^2 = \Delta$. In this case, we have the following lemma:

**Lemma A.3.** *Suppose that $f$ is a function where the eigenvalues of its Jacobian matrix have the following property:*

$$1 - 2\epsilon \leq \lambda_i \leq 1 + 2\epsilon, \ \forall i \in \{1, \ldots, d\}, \tag{80}$$

*and there exists some point $\boldsymbol{X}_0 \in \mathbb{R}^d$ where $\| \boldsymbol{X}_0 - f \left( \boldsymbol{X}_0 \right) \|_2 = \Delta$, then we have the followings:*

$$
\begin{aligned}
\| \mathbb{E} \left[ f \left( \boldsymbol{X} \right) - \boldsymbol{X} \right] \|_2 &\geq \Delta - 2\epsilon \mathbb{E} \left[ \| \boldsymbol{X} - \boldsymbol{X}_0 \|_2 \right], \\
\| f \left( \boldsymbol{X} \right) - \boldsymbol{X} \|_2 &\leq \Delta + 2R\epsilon, \ \forall \boldsymbol{X} : \| \boldsymbol{X} - \boldsymbol{X}_0 \|_2 \leq R, \\
\| f \left( \boldsymbol{X} \right) - \boldsymbol{X} \|_2 &\geq \Delta - 2R\epsilon, \ \forall \boldsymbol{X} : \| \boldsymbol{X} - \boldsymbol{X}_0 \|_2 \leq R.
\end{aligned} \tag{81}
$$

Now based on the result of Lemma A.3, if we have $\mathbb{E} \left[ \| \boldsymbol{X} - \boldsymbol{X}_0 \|_2 \right] \leq R$, then we have the followings:

$$\Delta - 2R\epsilon \leq \| \mathbb{E} \left[ f \left( \boldsymbol{X} \right) - \boldsymbol{X} \right] \|_2 \leq \max_{s \in \{+1, -1\}} \inf_{\mu \in \mathcal{C}(P^s, Q^s)} \mathbb{E} \left( \| \boldsymbol{X} - \boldsymbol{Y} \|_2 \right), \tag{82}$$

where $Q$ is the distribution of $\boldsymbol{Y} = f \left( \boldsymbol{X} \right)$, and $P^s = P \left( \boldsymbol{X} | y = s \right)$. On the other-hand due to Auxiliary lemma A.2, if $\lambda > \eta$ then we have:

$$\frac{1}{2} \max_{s \in \{+1, -1\}} \inf_{\mu \in \mathcal{C}(P^s, Q^s)} \mathbb{E} \left( \| \boldsymbol{X} - \boldsymbol{Y} \|_2 \right) \leq \inf_{\mu \in \mathcal{C}(P, Q)} \mathbb{E} \left( \| \boldsymbol{X} - \boldsymbol{Y} \|_2 \right) \leq \eta \tag{83}$$

Therefore we have :

$$\Delta \leq 2\eta + 4R\epsilon. \tag{84}$$

Now suppose that $h^s$ is the classifier with minimum standard error $\delta$. If we assume the region of the space where this classifier misclassify as $A$ and the distribution has the $(C_1, C_2) -$ expansion then we have :

$$
\begin{aligned}
\inf_{h \in \mathcal{H}} \sup_{P \in \mathcal{B}_\eta(P_0|\mathcal{F})} \mathbb{E}_P \left[ \ell \left( y, h \left( \boldsymbol{X} \right) \right) \right] &\leq P_0 \left( \mathcal{N}_{\Delta_{\max}} \left( A \right) \right) \\
&\leq \left( 1 + C_1 \Delta_{\max} \right) P \left( A \right) \\
&\leq \left( 1 + C_1 \left( 4R\epsilon + 2\eta \right) \right) \alpha.
\end{aligned} \tag{85}
$$

And the proof is complete. $\qquad\square$

---

[2] $\mathbf{LC} \left( P | \Delta, \mathcal{G} \right)$ is a function that changes the label of a set whose measure, according to $P$, is at most $\Delta$, so that the resulting measure falls into the $\mathcal{G}$ class

**Algorithm 3:** DRO-based Domain Adaptation For Expandable and Smooth Distributions

---

**Params:** $\Theta$, $\mathcal{G}$, $p$, $q$, $\lambda$, and $\eta$

**Input** : $P_0$, $\{P_{i_\mathcal{X}}\}_{1:T}$

**Initialize:**

$$\varepsilon_0 \longleftarrow \eta, \quad \widehat{P}_0 \longleftarrow P_0$$

$$\Delta_0^*, \theta_0^* \longleftarrow \left\{ \min_{\theta \in \Theta}, \operatorname*{argmin}_{\theta \in \Theta} \right\} \sup_{P \in \mathcal{B}_{\varepsilon_0}(P_0 | \mathcal{F})} \mathbb{E}_P\left[\ell\left(y, h_\theta\left(\boldsymbol{X}\right)\right)\right].$$

**for** $i = 1, \ldots, T-1$ **do**

$$\widetilde{P}_i \longleftarrow P_{i_\mathcal{X}}\left(\boldsymbol{X}\right) \mathbb{1}\left(y = h_{\theta_{i-1}^*}\left(\boldsymbol{X}\right)\right), \quad \forall \left(\boldsymbol{X}, y\right) \in \mathcal{Z}$$

$$\widehat{P}_i \longleftarrow \mathbf{LC}\left(\widetilde{P}_i \Big| \Delta_{i-1}^*, \mathcal{G}\right)^2$$

$$\varepsilon_i \longleftarrow 2\lambda \Delta_{i-1}^* + \eta$$

$$\Delta_i^*, \theta_i^* \longleftarrow \left\{ \min_{\theta \in \Theta}, \operatorname*{argmin}_{\theta \in \Theta} \right\} \sup_{P \in \mathcal{B}_{\varepsilon_i}(\widehat{P}_i | \mathcal{F})} \mathbb{E}_P\left[\ell\left(y, h_\theta\left(\boldsymbol{X}\right)\right)\right]$$

**Result:** $\theta^* \longleftarrow \theta_{T-1}^*$

---

*Proof of Theorem 4.6.* We show for each $\theta \in \Theta$, $R^{\mathrm{CDRL}}\left(\theta; \widehat{P}_0\right)$ is converging to $R^{\mathrm{CDRL}}\left(\theta; P_0\right)$. Assume $\mathcal{F}$ is the family of displacement functions; i.e. each $f \in \mathcal{F}$ is moving all the points within a fixed vector $\delta$. suppose we have $n$ empirical samples from $P_0$ named $z_1, z_2, \ldots, z_n$. Also Suppose $S^+$ contains indices of positive class samples and $S^-$ similarly for negative class samples. Then

$$R^{\mathrm{CDRL}}\left(\theta; \widehat{P}_0\right) = \sup_{f \in \mathcal{F}} \frac{1}{n} \sum_{i=1}^n \mathbb{1}\left(y_i \langle \theta, f_{y_i}(x_i) \rangle < 0\right) - \gamma c(x_i, f_{y_i}(x_i))$$

$$= \frac{1}{n} \sup_{\delta_1} \left( \sum_{i \in S^+} \mathbb{1}\left(\langle \theta, \delta_1 + x_i \rangle < 0\right) - \gamma \|\delta_1\|_2 \right)$$

$$+ \frac{1}{n} \sup_{\delta_2} \left( \sum_{i \in S^-} \mathbb{1}\left(\langle \theta, \delta_2 + x_i \rangle > 0\right) - \gamma \|\delta_2\|_2 \right)$$

Now just focus on positive class samples and we know the underlying distribution for each sample is according to Gaussian distribution with parameters $(\mu, \sigma^2 I)$. It is obvious that for each $\theta$, the supremum is maximized when $\delta$ is in the same direction as $\theta$. Then without loss of generality assume $\|\theta\|_2 = 1$ and define $m := |S^+|$ and $p_i := \langle x_i, \theta \rangle$. Also assume $m = \frac{n}{2}$ with high probability. Hence we have:

$$\frac{1}{m} \sup_{\delta} \left( \sum_{i \in S^+} \mathbb{1}\left(\langle \theta, \delta + x_i \rangle < 0\right) - \gamma \|\delta\|_2 \right) = \sup_{t \geq 0} \left( \frac{\#(p_i < t)}{m} - \gamma t \right) \tag{86}$$

Now if we consider $R^{\mathrm{CDRL}}\left(\theta; P_0\right)$ for positive class samples the above expression would become:

$$\sup_{t \geq 0} P_0[\langle \theta, X_i \rangle < t] - \gamma t$$

Note $\langle \theta, X_i \rangle$ is one dimensional Gaussian distribution with parameters $(\langle \theta, \mu \rangle, \sigma^2)$ and we denote it's CDF with $F_\theta(X)$. Using derivatives we have at the maximization point:

$$\gamma = \frac{1}{\sqrt{2\pi\sigma^2}} \exp\left(-\frac{(t - \langle \theta, \mu \rangle)^2}{2\sigma^2}\right)$$

Then the maximization point is:

$$t = \langle \theta, \mu \rangle + \sigma \sqrt{2 \log \frac{1}{\gamma \sqrt{2\pi\sigma^2}}}$$

Note $t = \langle \theta, \mu \rangle - \sigma \sqrt{2 \log \frac{1}{\gamma \sqrt{2 \pi \sigma^2}}}$ doesn't make the expression maximum because at that point increasing $t$ will increase the expression. Also note if $\gamma > \frac{1}{\sqrt{2 \pi \sigma^2}}$ the supremum doesn't exist which means the optimum case is not moving any point.

Now we can use the uniform convergence of $F_\theta(X)$ and $\widehat{F}_\theta(X)$. We have with probability at least $1 - \delta$:

$$\sup_{t \geq 0} \left| \frac{\#(p_i < t)}{m} - P_0[\langle \theta, X_i \rangle < t] \right| \leq 4 \sqrt{\frac{\log(m+1)}{m}} + \sqrt{\frac{2}{m} \log \frac{2}{\delta}} < 4 \sqrt{\frac{2}{m} \log \frac{4m}{\delta}}$$

Then for a fixed $\theta$, with probability at least $1 - \delta$ we conclude:

$$\left| \sup_{t \geq 0} \left( \frac{\#(\langle x_i, \theta \rangle < t)}{m} - \gamma t \right) - \sup_{t \geq 0} \left( P_0[\langle \theta, X_i \rangle < t] - \gamma t \right) \right| < 8 \sqrt{\frac{2}{m} \log \frac{4m}{\delta}}$$

Therefore we can conclude for a fixed $\theta$, with probability at least $1 - 2\delta$:

$$\left| R^{\mathrm{CDRL}}(\theta; P_0) - R^{\mathrm{CDRL}}\left(\theta; \widehat{P}_0\right) \right| < 32 \sqrt{\frac{1}{n} \log \frac{2n}{\delta}}$$

Now we can extend this bound for all $\theta \in \Theta$ via quantization on $\theta$. Assume that all of data points are inside a $R$-Ball with high probability. if $\|\theta_1 - \theta_2\| < \epsilon$, $\|\langle x_i, \theta_1 \rangle - \langle x_i, \theta_2 \rangle\| < \epsilon R$ and the close-form answer of $\sup_{t \geq 0} \left( P_0[\langle \theta, X_i \rangle < t] - \gamma t \right)$ differs at most $\mu \epsilon \leq \epsilon R$. Hence, with $\epsilon$-covering of $\Theta$ we conclude with probability $1 - 2\delta$ for each $\theta \in \Theta$:

$$\forall \theta \in \Theta : \left| R^{\mathrm{CDRL}}(\theta; P_0) - R^{\mathrm{CDRL}}\left(\theta; \widehat{P}_0\right) \right| < 4\epsilon R + 32 \sqrt{\frac{1}{n} \left( d \log \left(1 + \frac{2}{\epsilon}\right) + \log \frac{2n}{\delta} \right)}$$

$$< 64 \sqrt{\frac{d}{n} \log \left( \frac{R}{\delta} \sqrt{\frac{n^3}{d}} \right)}$$

$\square$

# B Proofs for the Lemmas

*proof of Lemma A.1.* To establish this theorem, we must show that if we have two Gaussian generative models, $P_{\boldsymbol{\mu}_1}$ and $P_{\boldsymbol{\mu}_2}$, with densities $f_1$ and $f_2$, where $\|\boldsymbol{\mu}_1 - \boldsymbol{\mu}_2\|_2 \geq \zeta$, and $\|\boldsymbol{\mu}_1 + \boldsymbol{\mu}_2\|_2 \geq \zeta$, then the Wasserstein distance between these two distributions has a non-zero lower bound. Building on the result of Lemma A.2, if we set $p = 2$ and $q = 1$, for any $\lambda \geq 0$, we have:

$$
\begin{aligned}
\mathcal{W}_c(f_1, f_2) &\geq \frac{1}{2} \min_{i,j \in \{-1,+1\}} \mathcal{W}_{2,\lambda}^1 \left( f_{1\boldsymbol{X}|y=i}, f_{2\boldsymbol{X}|y=j} \right) \\
&= \frac{1}{2} \min \left\{ \|\boldsymbol{\mu}_1 - \boldsymbol{\mu}_2\|_2, \|\boldsymbol{\mu}_1 + \boldsymbol{\mu}_2\|_2 \right\} \\
&\geq \frac{\zeta}{2},
\end{aligned}
\tag{87}
$$

This concludes the proof. $\square$

*Proof of Lemma A.2.* The Wasserstein distance between two distributions $P$ and $Q$ which are both supported over $\mathcal{Z}$ is defined as

$$\mathcal{W}_{\tilde{c}}(P, Q) \triangleq \inf_{\rho \in \Omega(P,Q)} \mathbb{E}_{(\boldsymbol{X}, y, \boldsymbol{X}', y') \sim \rho} \left( \tilde{c}(\boldsymbol{X}, y; \boldsymbol{X}', y') \right), \tag{88}$$

where $\Omega(P, Q)$ denotes the set of all couplings (i.e., joint distributions) on $\mathcal{Z}^2$ that have $P$ and $Q$ as their respective marginals. Based on the above definition, the distance between $P_{\boldsymbol{\mu}_1}$ and $P_{\boldsymbol{\mu}_2}$ can be

attained via the following formula:

$$\mathcal{W}_{\tilde{c}}\left(P_{\boldsymbol{\mu}_1}, P_{\boldsymbol{\mu}_2}\right) = \inf_{\rho \in \Omega(f_1, f_2)} \sum_{y, y'} \int \tilde{c}(\boldsymbol{X}, y; \boldsymbol{X}', y') \rho(\boldsymbol{X}, y, \boldsymbol{X}', y') \mathrm{d}\boldsymbol{X} \mathrm{d}\boldsymbol{X}' \tag{89}$$

$$= \inf_{\rho \in \Omega(f_1, f_2)} \frac{1}{2} \sum_{y'} \int \tilde{c}(\boldsymbol{X}, 1; \boldsymbol{X}', y') \rho(\boldsymbol{X}', y' | \boldsymbol{X}, y = 1) f_1(\boldsymbol{X} | y = 1) \mathrm{d}\boldsymbol{X} \mathrm{d}\boldsymbol{X}'$$

$$+ \frac{1}{2} \sum_{y'} \int \tilde{c}(\boldsymbol{X}, -1; \boldsymbol{X}', y') \rho(\boldsymbol{X}', y' | \boldsymbol{X}, y = -1) f_1(\boldsymbol{X} | y = -1) \mathrm{d}\boldsymbol{X} \mathrm{d}\boldsymbol{X}',$$

where due to the definition of $P_{\boldsymbol{\mu}_1}$, we have that $f_1(\boldsymbol{X}|y)$ is the density function of a Gaussian distribution with mean $y\boldsymbol{\mu}_1$ and covariance matrix $\sigma^2 I$. Regarding the density function $\rho$ in equation (89), we have the following set of constraints:

$$i) \quad f_2(\boldsymbol{X}', y') = \frac{1}{2} \int \rho(\boldsymbol{X}', y' | \boldsymbol{X}, y = 1) f_1(\boldsymbol{X} | y = 1) \mathrm{d}\boldsymbol{X}$$

$$+ \frac{1}{2} \int \rho(\boldsymbol{X}', y' | \boldsymbol{X}, y = -1) f_1(\boldsymbol{X} | y = -1) \mathrm{d}\boldsymbol{X}, \quad \forall \boldsymbol{X}', y'. \tag{90}$$

$$ii) \quad \sum_{y' \in \{\pm 1\}} \int \rho(\boldsymbol{X}', y' | \boldsymbol{X}, y = 1) \mathrm{d}\boldsymbol{X}' = 1, \quad \forall \boldsymbol{X}. \tag{91}$$

$$iii) \quad \sum_{y' \in \{\pm 1\}} \int \rho(\boldsymbol{X}', y' | \boldsymbol{X}, y = -1) \mathrm{d}\boldsymbol{X}' = 1, \quad \forall \boldsymbol{X}. \tag{92}$$

Let us define a non-negative function $\tilde{\rho}$ as follows:

$$\tilde{\rho}(\boldsymbol{X}, \boldsymbol{X}', y') \triangleq \frac{1}{2} \frac{\rho(\boldsymbol{X}', y' | \boldsymbol{X}, y = -1) f_1(\boldsymbol{X} | y = -1)}{f_1(\boldsymbol{X} | y = 1)} + \frac{1}{2} \rho(\boldsymbol{X}', y' | \boldsymbol{X}, y = 1). \tag{93}$$

It should be noted that $\tilde{\rho}$ may not even be a *probability density* since it may not integrate into one over all possible values of $\boldsymbol{X}, \boldsymbol{X}'$ and $y'$. In any case, for this function (i.e., $\tilde{\rho}$), we have:

$$(*) \quad \int \tilde{\rho}(\boldsymbol{X}, \boldsymbol{X}', y') f_1(\boldsymbol{X} | y = 1) \mathrm{d}\boldsymbol{X} = f_2(\boldsymbol{X}', y'), \quad \forall \boldsymbol{X}', y'. \tag{94}$$

$$(**) \quad \sum_{y' \in \{\pm 1\}} \int \tilde{\rho}(\boldsymbol{X}, \boldsymbol{X}', y') \mathrm{d}\boldsymbol{X}' = \frac{1}{2} \left( 1 + \frac{f_1(\boldsymbol{X} | y = 1)}{f_1(\boldsymbol{X} | y = -1)} \right), \quad \forall \boldsymbol{X}. \tag{95}$$

Therefore, if there exists a joint density $\rho$ that satisfies the set of constraints i), ii) and iii) (respectively defined in (90), (91), and (92)), then there also exists a function $\tilde{\rho}$ that satisfies the set of constraints in (*) and (**) as defined in (94) and (95), respectively. Additionally, since we know that $f_1$ is a non-negative function, we can further relax the conditions as follows:

$$\int \tilde{\rho}(\boldsymbol{X}, \boldsymbol{X}', y') f_1(\boldsymbol{X} | y = 1) \mathrm{d}\boldsymbol{X} \geq f_2(\boldsymbol{X}', y'), \quad \forall \boldsymbol{X}', y'. \tag{96}$$

$$\sum_{y' \in \{\pm 1\}} \int \tilde{\rho}(\boldsymbol{X}, \boldsymbol{X}', y') \mathrm{d}\boldsymbol{X}' \geq \frac{1}{2}, \quad \forall \boldsymbol{X}. \tag{97}$$

Therefore, the constraints in (96) and (97) are relaxed versions of the constraints in (90), (91), and (92). Let us denote the set of all non-negative functions $\tilde{\rho}$ that satisfy the (newer versions of)

conditions (*) and (**) with $\Pi$. Then, we have:

$$
\begin{aligned}
\mathcal{W}_{\tilde{c}} & \left(P_{\boldsymbol{\mu}_1}, P_{\boldsymbol{\mu}_2}\right) \\
= & \inf_{\rho \in \Omega(f_1, f_2)} \frac{1}{2} \sum_{y'} \int \tilde{c}(\boldsymbol{X}, 1; \boldsymbol{X}', y') \rho(\boldsymbol{X}', y'|\boldsymbol{X}, y = 1) f_1(\boldsymbol{X}|y = 1) \mathrm{d}\boldsymbol{X}\mathrm{d}\boldsymbol{X}' \\
& + \frac{1}{2} \sum_{y'} \int \tilde{c}(\boldsymbol{X}, -1; \boldsymbol{X}', y') \rho(\boldsymbol{X}', y'|\boldsymbol{X}, y = -1) f_1(\boldsymbol{X}|y = -1) \mathrm{d}\boldsymbol{X}\mathrm{d}\boldsymbol{X}' \\
= & \inf_{\rho \in \Omega(f_1, f_2)} \frac{1}{2} \sum_{y'} \int c(\boldsymbol{X}, \boldsymbol{X}') \rho(\boldsymbol{X}', y'|\boldsymbol{X}, y = 1) f_1(\boldsymbol{X}|y = 1) \mathrm{d}\boldsymbol{X}\mathrm{d}\boldsymbol{X}' \\
& + \frac{1}{2} \sum_{y'} \int c(\boldsymbol{X}, \boldsymbol{X}') \rho(\boldsymbol{X}', y'|\boldsymbol{X}, y = -1) f_1(\boldsymbol{X}|y = -1) \mathrm{d}\boldsymbol{X}\mathrm{d}\boldsymbol{X}' \\
& + \frac{\lambda}{2} \left(\rho(y' = -1|y = 1) + \rho(y' = 1|y = -1)\right) \\
\geq & \inf_{\rho \in \Omega(f_1, f_2)} \frac{1}{2} \sum_{y'} \int c(\boldsymbol{X}, \boldsymbol{X}') \rho(\boldsymbol{X}', y'|\boldsymbol{X}, y = 1) f_1(\boldsymbol{X}|y = 1) \mathrm{d}\boldsymbol{X}\mathrm{d}\boldsymbol{X}' \\
& + \frac{1}{2} \sum_{y'} \int c(\boldsymbol{X}, \boldsymbol{X}') \rho(\boldsymbol{X}', y'|\boldsymbol{X}, y = -1) f_1(\boldsymbol{X}|y = -1) \mathrm{d}\boldsymbol{X}\mathrm{d}\boldsymbol{X}',
\end{aligned}
$$

which due to the definition of $\Pi$ and its discussed properties imply the following bound on the Wasserstein distance between $f_1$ and $f_2$:

$$
\begin{aligned}
\mathcal{W}_{\tilde{c}} \left(P_{\boldsymbol{\mu}_1}, P_{\boldsymbol{\mu}_2}\right) & \geq \widehat{\mathcal{W}}_c \left(f_1, f_2\right) \\
& \triangleq \inf_{\tilde{\rho} \in \Pi} \sum_{y'} \int c(\boldsymbol{X}, \boldsymbol{X}') \tilde{\rho}(\boldsymbol{X}, \boldsymbol{X}', y') f_1(\boldsymbol{X}|y = 1) \mathrm{d}\boldsymbol{X}\mathrm{d}\boldsymbol{X}' \\
& = \inf_{\tilde{\rho} \in \Pi} \int c(\boldsymbol{X}, \boldsymbol{X}') \left[\sum_{y'} \tilde{\rho}(\boldsymbol{X}, \boldsymbol{X}', y')\right] f_1(\boldsymbol{X}|y = 1) \mathrm{d}\boldsymbol{X}\mathrm{d}\boldsymbol{X}' \\
& = \inf_{\tilde{\rho} \in \Pi} \int c(\boldsymbol{X}, \boldsymbol{X}') \tilde{\rho}(\boldsymbol{X}, \boldsymbol{X}') f_1(\boldsymbol{X}|y = 1) \mathrm{d}\boldsymbol{X}\mathrm{d}\boldsymbol{X}', \quad (98)
\end{aligned}
$$

where $\tilde{\rho}(\boldsymbol{X}, \boldsymbol{X}')$ is defined as

$$
\tilde{\rho}(\boldsymbol{X}, \boldsymbol{X}') \triangleq \sum_{y'} \tilde{\rho}(\boldsymbol{X}, \boldsymbol{X}', y'). \quad (99)
$$

The rest of the proof proceeds by trying to find a proper structure for $\Pi$. In order to do so, let us define $\Pi^{\oplus}$ as the following set:

$$
\Pi^{\oplus} \triangleq \left\{\sum_{i \in \{1,2\}} \sum_{y' \in \{\pm 1\}} \tilde{\rho}_i\left(\cdot, \cdot, y'\right) \,\middle|\, \tilde{\rho}_1, \tilde{\rho}_2 \in \Pi\right\} \subseteq \mathbb{R}_+^{\mathcal{X} \times \mathcal{X}}. \quad (100)
$$

Then, it can be readily seen that the following bound can be established:

$$
\begin{aligned}
\mathcal{W}_{\tilde{c}} \left(P_{\boldsymbol{\mu}_1}, P_{\boldsymbol{\mu}_2}\right) & \geq \widehat{\mathcal{W}}_c \left(f_1, f_2\right) \\
& \geq \frac{1}{2} \inf_{\zeta \in \Pi^{\oplus}} \int c(\boldsymbol{X}, \boldsymbol{X}') \zeta(\boldsymbol{X}, \boldsymbol{X}') f_1(\boldsymbol{X}|y = 1) \mathrm{d}\boldsymbol{X}\mathrm{d}\boldsymbol{X}'. \quad (101)
\end{aligned}
$$

Also, we should keep in mind that each $\zeta$ in $\Pi^\oplus$ has the following properties:

$$\forall \zeta \in \Pi^\oplus \to \exists \tilde{\rho}_1, \tilde{\rho}_2 \in \Pi : \zeta\left(\boldsymbol{X}, \boldsymbol{X}'\right) = \sum_{y' \in \{\pm 1\}} \sum_{i=1,2} \tilde{\rho}_i\left(\boldsymbol{X}, \boldsymbol{X}', y'\right), \forall \boldsymbol{X}, \boldsymbol{X}', \quad (102)$$

$i)$ $\int \zeta\left(\boldsymbol{X}, \boldsymbol{X}'\right) f_1(\boldsymbol{X}|y = 1)\mathrm{d}\boldsymbol{X}$

$$= \sum_{y' \in \{\pm 1\}} \int \left[\tilde{\rho}_1\left(\boldsymbol{X}, \boldsymbol{X}', y'\right) + \tilde{\rho}_2\left(\boldsymbol{X}, \boldsymbol{X}', y'\right)\right] f_1(\boldsymbol{X}|y = 1)\mathrm{d}\boldsymbol{X} = 2f_2\left(\boldsymbol{X}'\right), \quad \forall \boldsymbol{X}',$$

$ii)$ $\int \zeta\left(\boldsymbol{X}, \boldsymbol{X}'\right)\mathrm{d}\boldsymbol{X}'$

$$= \sum_{y' \in \{\pm 1\}} \int \left[\tilde{\rho}_1\left(\boldsymbol{X}, \boldsymbol{X}', y'\right) + \tilde{\rho}_2\left(\boldsymbol{X}, \boldsymbol{X}', y'\right)\right]\mathrm{d}\boldsymbol{X}' \geq 1, \quad \forall \boldsymbol{X},$$

which hold due to (96) and (97). Now, we define two more sets, denoted by $\Pi_-, \Pi_+ \subseteq \mathbb{R}^{\mathcal{X} \times \mathcal{X}}$ according to the following definitions. For $s \in \{\pm\}$, let us define:

$$\forall \xi \in \Pi_s : \exists \tilde{\rho} \in \Pi \to \xi\left(\boldsymbol{X}, \boldsymbol{X}'\right) = \sum_{y' \in \{\pm 1\}} \tilde{\rho}\left(\boldsymbol{X}, \boldsymbol{X}', y'\right),$$

$$\sum_{y'' \in \{\pm 1\}} \int \tilde{\rho}(\boldsymbol{X}, \boldsymbol{X}', y'') f_1(\boldsymbol{X}|y = 1)\mathrm{d}\boldsymbol{X} \geq f_2(\boldsymbol{X}'|y' = s), \quad \forall \boldsymbol{X}',$$

$$\sum_{y' \in \{\pm 1\}} \int \tilde{\rho}(\boldsymbol{X}, \boldsymbol{X}', y')\mathrm{d}\boldsymbol{X}' \geq 1, \quad \forall \boldsymbol{X}. \quad (103)$$

What remains to do is to show that for any $\zeta$ in $\Pi^\oplus$, there exists at least a pair $(\xi_-, \xi_+) \in \Pi_- \times \Pi_+$ such that $\zeta \geq (\xi_- + \xi_+)/2$ everywhere in $\mathcal{X}^2$. This can be easily verified by seeing that since we have:

$$2f_2\left(\boldsymbol{X}'\right) = f_2\left(\boldsymbol{X}'|y' = 1\right) + f_2\left(\boldsymbol{X}'|y' = -1\right), \quad (104)$$

the constraints for $\zeta \in \Pi^\oplus$ which are derived in (102) always hold for average between any two members of the due $(\Pi_-, \Pi_+)$. Therefore, we can further bound the Wasserstein distance between $f_1$ and $f_2$ via the following chain of inequalities:

$$\mathcal{W}_{\tilde{c}}\left(P_{\boldsymbol{\mu}_1}, P_{\boldsymbol{\mu}_2}\right) \geq \widehat{\mathcal{W}}_c\left(f_1, f_2\right) \quad (105)$$

$$\triangleq \inf_{\tilde{\rho} \in \Pi} \int c(\boldsymbol{X}, \boldsymbol{X}')\tilde{\rho}(\boldsymbol{X}, \boldsymbol{X}')f_1(\boldsymbol{X}|y = 1)\mathrm{d}\boldsymbol{X}\mathrm{d}\boldsymbol{X}'$$

$$\geq \frac{1}{2} \inf_{\zeta \in \Pi^\oplus} \int c(\boldsymbol{X}, \boldsymbol{X}')\zeta(\boldsymbol{X}, \boldsymbol{X}')f_1(\boldsymbol{X}|y = 1)\mathrm{d}\boldsymbol{X}\mathrm{d}\boldsymbol{X}'$$

$$\geq \frac{1}{2} \inf_{\xi_\pm \in (\Pi_\pm)} \int c(\boldsymbol{X}, \boldsymbol{X}') \left[\frac{\xi_+(\boldsymbol{X}, \boldsymbol{X}') + \xi_-(\boldsymbol{X}, \boldsymbol{X}')}{2}\right] f_1(\boldsymbol{X}|y = 1)\mathrm{d}\boldsymbol{X}\mathrm{d}\boldsymbol{X}'$$

$$\geq \frac{1}{2} \min_{s \in \{\pm 1\}} \inf_{\xi \in \Pi_s} \int c(\boldsymbol{X}, \boldsymbol{X}')\xi_s(\boldsymbol{X}, \boldsymbol{X}')f_1(\boldsymbol{X}|y = 1)\mathrm{d}\boldsymbol{X}\mathrm{d}\boldsymbol{X}'.$$

For any $s \in \{\pm 1\}$, $\xi_s\left(\boldsymbol{X}', \boldsymbol{X}\right)$ acts as a surrogate for $\rho\left(\boldsymbol{X}'|\boldsymbol{X}, y = 1, y' = s\right)$ where $\rho \in \Omega\left(f_1, f_2\right)$. However, the *marginal* and *normalization* equality constraints, i.e.,

$$\int \rho\left(\boldsymbol{X}'|\boldsymbol{X}, y = 1, y' = s\right) f_1(\boldsymbol{X}|y = 1)\mathrm{d}\boldsymbol{X} = f_2\left(\boldsymbol{X}'|y' = s\right),$$

$$\int \rho\left(\boldsymbol{X}'|\boldsymbol{X}, y = 1, y' = s\right) = 1, \quad (106)$$

have been relaxed and, in fact, replaced by inequalities. However, since the optimization problem $\inf_{\xi \in \Pi_s}$ is a linear program with both linear objective and constraints, the optimal point (if exists) always occurs at the boundaries of the feasible set where constraints are active. The objective is

non-negative and thus bounded below, thus the optimal point exists. On the other hand, neither of the constraints are degenerate and hence they all become active. Therefore, we have

$$\inf_{\xi \in \Pi_s} \int c(\boldsymbol{X}, \boldsymbol{X}')\xi_s(\boldsymbol{X}, \boldsymbol{X}')f_1(\boldsymbol{X}|y=1)\mathrm{d}\boldsymbol{X}\,\mathrm{d}\boldsymbol{X}' = \mathcal{W}_c\left(f_1\left(\cdot|y=1\right), f_2\left(\cdot|y'=s\right)\right), \quad (107)$$

and as a result, we have

$$\mathcal{W}_{\tilde{c}}\left(f_1, f_2\right)$$
$$\geq \frac{1}{2}\min\left\{\mathcal{W}_c\left(f_1(\cdot|y=1), f_2\left(\cdot|y'=1\right)\right), \ \mathcal{W}_c\left(f_1(\cdot|y=1), f_2\left(\cdot|y'=-1\right)\right)\right\}. \quad (108)$$

Also, it should be noted that the whole proof can be re-written from the start with $f_1(\cdot|y=-1)$ instead of conditioning on $y=1$. Therefore, the final bound can be written as

$$\mathcal{W}_{\tilde{c}}\left(f_1, f_2\right) \geq \frac{1}{2}\max_{i \in \{\pm 1\}}\min_{j \in \{\pm 1\}} \mathcal{W}_c\left(f_1\left(\cdot|y=i\right), f_2\left(\cdot|y'=j\right)\right), \quad (109)$$

which completes the proof. $\qquad\square$

*Proof of Lemma A.3.* We write the Mean value theorem for the function $f$ around $\boldsymbol{X}_0$ we have:

$$f\left(\boldsymbol{X}\right) = f\left(\boldsymbol{X}_0\right) + \boldsymbol{J}_f\left(\boldsymbol{X}'\right)\left(\boldsymbol{X} - \boldsymbol{X}_0\right), \quad (110)$$

where $\boldsymbol{J}_f\left(\boldsymbol{X}'\right)$ is the Jacobian matrix of $f$ in $\boldsymbol{X}'$, and $\boldsymbol{X}'$ is a point between $\boldsymbol{X}_0$ and $\boldsymbol{X}$. From the properties of $\boldsymbol{J}_f$ in the lemma, we can continue the above inequalities as follows:

$$f\left(\boldsymbol{X}\right) - \boldsymbol{X} = f\left(\boldsymbol{X}_0\right) - \boldsymbol{X_0} + \left(\boldsymbol{J}_f\left(\boldsymbol{X}'\right) - \boldsymbol{I}_d\right)\left(\boldsymbol{X} - \boldsymbol{X}_0\right), \quad (111)$$

where $\boldsymbol{I}_d$ is the $d \times d$ identity matrix. Now we have:

$$\|\mathbb{E}\left[f\left(\boldsymbol{X}\right) - \boldsymbol{X}\right]\|_2 \geq \|f\left(\boldsymbol{X}_0\right) - \boldsymbol{X_0}\|_2 - \|\mathbb{E}\left[\left(\boldsymbol{J}_f\left(\boldsymbol{X}'\right) - \boldsymbol{I}_d\right)\left(\boldsymbol{X} - \boldsymbol{X}_0\right)\right]\|_2$$
$$\geq \Delta - 2\epsilon\mathbb{E}\left[\|\boldsymbol{X} - \boldsymbol{X}_0\|_2\right]. \quad (112)$$

We also can continue equation 111 as follows:

$$\|f\left(\boldsymbol{X}\right) - \boldsymbol{X}\|_2 \leq \|f\left(\boldsymbol{X}_0\right) - \boldsymbol{X_0}\|_2 + 2\epsilon\|\boldsymbol{X} - \boldsymbol{X}_0\|_2$$
$$\leq \Delta + 2R\epsilon,$$
$$\|f\left(\boldsymbol{X}\right) - \boldsymbol{X}\|_2 \geq \|f\left(\boldsymbol{X}_0\right) - \boldsymbol{X_0}\|_2 - 2\epsilon\|\boldsymbol{X} - \boldsymbol{X}_0\|_2$$
$$\geq \Delta - 2R\epsilon \quad (113)$$

Which completes the proof. $\qquad\square$

