# OpenReview forum: "Gradual Domain Adaptation via Manifold-Constrained Distributionally Robust Optimization"
_NeurIPS.cc/2024/Conference — NeurIPS 2024 poster_

### Official Review · Reviewer_1exJ · 2024-07-07

**Soundness:** 3
**Presentation:** 3
**Contribution:** 2
**Rating:** 6
**Confidence:** 3

**Summary:**

This paper introduces a novel approach to gradual domain adaptation using distributionally robust optimization (DRO). The core idea is to adapt models across successive datasets by controlling the Wasserstein distance between distributions and ensuring they lie on a favorable manifold.  The authors  apply  the method  theoretically to two examples and  provide theoretical  guarantees for generalization error. Furthermore, the authors also validate the  theoretical findings through a series of experiments.

**Strengths:**

+ The theoretical contributions are substantial, providing rigorous guarantees on model adaptation and generalization errors across domains.  The algorithm provides a  bounded error regardless of  $T$​,  For appropriately constrained  distributions, the error can be demonstrated to be linear or even entirely eradicated.

+  This paper extend the theoretical results to   a more general class of distributions (referred  to as "expandable" distributions)  with learnable classifiers.
+ Furthermore,  The authors demonstrate the polynomial-time convergence of the algorithm.

* The paper is well-written and structured, effectively communicating complex theoretical concepts in a clear and organized manner. It employs rigorous mathematical formalism while maintaining readability, making it accessible to readers

**Weaknesses:**

1.  It seems that the proof of the theoretical results requires overly stringent conditions, e.g., the assumption of expandable distributions,  the use of smooth mappings and the requirement of   distribution  characterized by  favorable properties.  And it  might not generalize well to real-world data distributions encountered in practice.

2. Both  Gaussian mixture model data distributions  and  "expandable" distributions are relatively toy data distributions,  which are somewhat different from the data distribution of real tasks.

3. The paper primarily focuses on the error within the domain $P_T$. However, the error of $\theta^*$ in the previous domains $P_1, \cdots, P_{T-1}$ remains uncertain.

4.  How to effectively compute the following WDRO problem **under the manifold constraint** $G$**?

$\Delta_i^*, \theta_i^* \longleftarrow \{\min _{\theta \in \Theta}, \underset{\theta \in \Theta}{\operatorname{argmin}}\} \sup *{P \in \mathcal{B}*{\varepsilon_i}\left(\widehat{P}_i \mid \mathcal{G}\right)} \mathbb{E}*P\left[\ell\left(y, h*\theta(\boldsymbol{X})\right)\right].$

5. This is related to the theoretical analysis of the Self-Training method in a semi-supervised scenario, where pseudo-labels are assigned to unlabeled data for training [1]. It seems to extend the transformation from $P_{labeled} \rightarrow P_{unlabel}$ ,  in this scenario to  $P_{labeled}^1 \rightarrow P_{unlabel}^2\rightarrow P_{unlabel}^3\rightarrow P_{unlabel}^4 ...\rightarrow P_{unlabel}^T$ and uses WDRO to capture the changes in Domain $P^1$ to $P^T$ (in Definition 4.1-4.3). The authors should discuss more about the Assumptions.

6. When estimate the WDRO radius $\epsilon_i \longleftarrow \lambda \Delta_{i-1}^*+\eta$. It appears that the $\Delta_i^*$ is meaningful only if the loss $\ell$ is the zero-one loss. This is because the definition of $\mathcal{W}_{p, \lambda}^q(P, Q)$  (as shown in Eq.(2) in the paper) is related to the zero-one loss $\mathbb{1}\left\{y \neq y^{\prime}\right\}$.

[1] Colin Wei, Kendrick Shen, Yining Chen, and Tengyu Ma. Theoretical analysis of self-training with deep networks on unlabeled data. In *International Conference on Learning Representations*, 2020.

**Questions:**

+ My main concern is that theoretical results rely on relatively toy data distributions and stringent properties, which could lead to limited practical applicability.

+ What are the challenges in extending theoretical results to multi-class data distributions?

**Limitations:**

see weaknesses.

---

> ### Author Rebuttal · Authors · 2024-08-04
>
> We would like to thank the reviewer for their feedback. The reviewer's main concern is the applicability of our method in real-world tasks which we have addressed in the camera-ready version. Below, we provide detailed responses to each of the reviewer's comments and concerns:
>
> ----
>
> **Weaknesses**:
>
> - **The proof of the theoretical results require stringent conditions, such as the assumption of expandability, the use of smooth mappings, and the requirement for distributions characterized by favorable properties. This may limit the generalizability of the results to real-world data distributions.**: These assumptions and toy examples were introduced to facilitate the theoretical analysis of our algorithm. However, the algorithm is robust and can be applied to real-world data without issues. Please refer to the global rebuttal and the attached PDF for more details, including results on real-world datasets commonly used in gradual domain adaptation literature, where our method outperforms competitors.
>
> - **Both GMM and "expandable" distributions are relatively toy examples, which are different from the real data distributions.**: As mentioned in our previous response, we have tested our algorithm on real-world datasets, and the results are available in the global response. These results will be included in the camera-ready version. Furthermore, our notion of expandability is closely related to the one used in previous work [WSCM20], where it has already been validated on datasets such as CIFAR-100.
>
> - **The paper focuses on the error within the domain $P_T$. However, the error of $\theta_T$ in the previous domains $P_1, ...,P_{{T-1}}$ remains uncertain.**: In our approach, we identify a $\theta^{\*}_i$ for each domain $P_i$ ($i \geq 1$) and provide guarantees for the error associated with $\theta^{\*}_i$ within its respective domain. $\theta^{\*}_T$ is specifically optimized for $P_T$, as each prior distribution has its own optimal solution.
>
> - **How to compute the WDRO problem under the manifold constraint of $\mathcal{G}$?**:
> We considered two scenarios: In the first, $\mathcal{G}$ is the family of Gaussian distributions, which is parameterizable, meaning we only need to tune the parameters while remaining on the manifold. This simplifies the problem to finding $P = P_\mu$, where $\Vert \mu - \mu_i \Vert_2 \leq \varepsilon_i$. In the second scenario, we assume that distributions can be generated from one another by applying an unknown but sufficiently smooth function. For example, $P_{i+1}$ models the distribution of $f(X)$, where $X \sim P_i$ and $f$ belongs to a general but smooth function family $\mathcal{F}$ (this condition is not overly restrictive in practice). In this case, we replace $P \in \mathcal{B}_{\varepsilon_i}(\widehat{P}_i | \mathcal{G})$
>
> with $P = f_{\\#} P_i$ for $f \in \mathcal{F}$. Practically, we consider a parametric family of functions for $\mathcal{F}$ and maximize the loss within this class using gradient ascent, while penalizing the function to ensure it does not deviate significantly from real samples.
>
>
> - **Regarding the analysis of Self-Training method in a semi-supervised scenario, where pseudo-labels are assigned to unlabeled data [WSCM20]. It seems to extend the transformation from $P_{l} \to P_{ul}$, in this scenario to $P^1_{l} \to P^2_{ul}  \to \ldots \to P^T_{ul} $and uses WDRO to capture the changes in Domain $P^1$ to $P^T$ (in Definition 4.1-4.3). The authors should discuss more about the Assumptions.**: Our work differs from [WSCM20] in several key aspects. In [WSCM20], the authors assume the availability of two potentially distant distributions over the feature space, with an unknown but shared labeling rule. They also assume access to a pseudo-labeler from the first distribution, which can be used to assign labels to unlabeled data from the second distribution. Additionally, it is assumed that both distributions and the unknown ground truth labeler satisfy certain robustness properties. In contrast, our work focuses on gradual domain adaptation, where consecutive distributions are assumed to be close. However, we make no assumptions about the labeling rule or the availability of the true distributions.
>
>
> - **When estimate the WDRO radius, it appears that the $ \Delta^{*}$ is meaningful only if the loss $ \ell $ is the zero-one loss. This is because the definition of distance (as shown in Eq.(2) in the paper) is related to the zero-one loss ${1}\\{y \neq y^{\prime}\\}$.**: The reviewer is correct that this result is directly applicable to the zero-one loss. However, it can be generalized to cases where $\ell$ or a scaled version of it, such as $\alpha \ell$, is greater than the (0,1)-loss. In this scenario, the estimation would be modified to $\epsilon_i \longleftarrow \alpha \lambda \Delta^*_{i-1} + \eta$. We will discuss this generalization in the camera-ready version.
>
> ----
>
> **Questions**:
>
> - **My main concern is that theoretical results rely on relatively toy data distributions and stringent properties, which could lead to limited practical applicability.** To address this concern, we implemented our method on a number of real-world datasets and achieved superior results compared to existing methods. Please refer to the global rebuttal for detailed results.
>
> - **What are the challenges in extending theoretical results to multi-class data distributions?**: As we mentioned in our response to Reviewer FbyN, our method, including Theorem 2.3 and Corollary 2.4, can be extended to multi-class settings. However, extending the examples involving Gaussian distributions and manifold assumptions would require additional statistical analysis, which we plan to address in future work. This will be discussed further in the camera-ready version.
>
> ----
>
> We hope these revisions address the majority of the reviewer’s concerns and lead to a favorable reassessment. Once again, thank you for your timely and thorough review of our work.

---

> > ### Comment · Reviewer_1exJ · 2024-08-12
> > **Appreciate**
> >
> > I appreciate the author's responses, and carefully read the comments from other reviewers. I agree with most of the reviewers that the paper presents a nice work. So I would like to raise my grade.

---

### Official Review · Reviewer_C19w · 2024-07-09

**Soundness:** 3
**Presentation:** 3
**Contribution:** 2
**Rating:** 6
**Confidence:** 2

**Summary:**

This paper studies the theoretical aspect of gradual domain adaptation (GDA), where the knowledge of labeled source domains is supposed to be transferred to a sequence of target domains. The main results show that the gradual adaptation process can be well characterized by the distributionally robust optimization (DRO) framework, where the domain gaps between the source domain and multiple target domains are gradually captured by the robustness of the model within a pre-set region, i.e., a ball w.r.t. Wasserstein metric over probability space. Finally, the distribution shift is guaranteed to be mitigated with the DRO algorithm.

**Strengths:**

+ The motivation of employing DRO as a theoretical framework to address gradual domain adaptation is clear and reasonable.
+ The extensive theoretical results seem to be solid.

**Weaknesses:**

- The clarity should be improved, where the advantages or improvements w.r.t. related theory for GDA is not discussed, which leads to unclear contributions.
- The presentation should be improved, e.g., the DRO algorithm is provided without justification while the main results closely depend on this algorithm.
- There are many typos and the readability is fair.

**Questions:**

Q1. As discussed in the previous work section, there are already several theoretical works for GDA, e.g., [WLZ22] and [HWLZ23]. However, the differences between the derived results and these works are not discussed in either *previous work section* or *main result section*. It should be clarified that what new insights are provided in the derived results.

Q2. The basic framework, i.e., DRO, is presented in Algorithm 1 directly, while there are no insights provided for it. Though there are plenty of results derived based on DRO, it is hard to understand the working mechanism of the DRO algorithm.

Q3. The bond in Theorem 2.3 shows that the target error can be dominated by the source risk with the factor $g_\lambda (\cdot)^{\circ T}$. Though Corollary 2.4 provides an analytic bound for $g_\lambda$, it would be more interesting to show the monotonicity of $g$ w.r.t. the composition operator. Furthermore, can the factor $g_\lambda (\cdot)^{\circ T}$ can be monotonically reduced by the increase of $T$? i.e., the factor of error can be reduced with the gradual adaptation process.

Q4. In literature [WLZ22], the main result of Theorem 1 shows that the target error is bounded by the source risk and accumulated error w.r.t. domain number $T$, which is induced by gradual adaptation process. Note that the main result in submission show the accumulated error is a factor of source risk, which implies this result could be loose when the factor is large. Some comparison between the tightness of these bounds are highly appreciated.


Minor: 1) Line 126, reference of Theorem; 2) Line 140, notations $\theta$ and $\Theta$ are used without definitions; 3) Line 186, previous studies [].

**Limitations:**

The theoretical analysis for gradual domain adaptation seems to have no potential negative societal impact.

---

> ### Author Rebuttal · Authors · 2024-08-04
>
> We would like to thank the reviewer for their feedback. The reviewer's main concern was the lack of sufficient discussion on certain aspects of the paper, which we have addressed in the revised/camera-ready version. Below, we provide detailed responses to each of the reviewer's comments and concerns:
>
> **Weaknesses**:
>
> - **Clarity needs improvement, especially in discussing the advantages or improvements related to GDA theory, leading to unclear contributions.**: We have enhanced the clarity and presentation of our work in the revised version. To provide a brief summary: as noted by other reviewers, our method has a significant advantage over GDA and other competing methods, where the error propagation in GDA grows exponentially with the number of intermediate domains $T$. Recent works [WLZ22, HWLZ23] have achieved linear error propagation, although the rate of growth remains constant even when the initial error is small. In contrast, our method can control error propagation to be independent of $T$, given certain assumptions on the distributions.
>
> - **The presentation should be improved, e.g., the DRO algorithm is provided without justification, despite the main results relying on this algorithm.**: We would like to clarify that DRO is a well-established technique and is not our original contribution. It has been extensively studied (see [1]) and previously applied in domain adaptation [2]. In our work, we propose adaptive robustness radii, and then utilize DRO by constraining the Wasserstein ball to a specific manifold of distributions, and analyze its performance in the context of gradual domain adaptation, both theoretically and experimentally (please refer to the global rebuttal for details on the new experiments).
>
> [1] Sinha et al., (2018). Certifying Some Distributional Robustness with Principled Adversarial Training. In International Conference on Learning Representations.
>
> [2] Lee et al. (2018). Minimax statistical learning with wasserstein distances. Advances in Neural Information Processing Systems, 31.
>
> - **Typos...**: We have carefully revised the paper to address this issue and assure the reviewer that all typos and rushed phrasing have been corrected.
>
> -------------------
>
> **Questions**:
>
> - **There are several theoretical works on GDA, e.g., [WLZ22] and [HWLZ23]. However, the differences between these works and the results in this paper are not discussed. It should be clarified what new insights are provided by the derived results.**: This question is addressed in our response to the first weakness. We will also highlight our contributions and the advantages of our method over previous methods in a dedicated subsection in the camera-ready version.
>
> - **The basic framework, i.e., DRO, is presented in Algorithm 1 directly, with no insights provided for it. Though there are plenty of results derived based on DRO, it is hard to understand the working mechanism of the DRO algorithm.**:  Theorem 1 (main theorem) demonstrates that using our DRO-based Algorithm 1, the error propagation can be uniquely and solely determined by a newly introduced complexity measure, $g_{\lambda}(\cdot)$. The rest of the paper is dedicated to obtaining $g(\cdot)$ for various settings and showing how error propagation can be entirely mitigated.
>
> - **Theorem 2.3 shows that the target error can be dominated by a factor of $g_{\lambda}(.)^{oT}$. Corollary 2.4 provides an analytic bound for $g_{\lambda}$, but it would be more interesting to show the monotonicity of $g$ with respect to the composition operator. Furthermore, can the factor $g_{\lambda}(.)^{oT}$ be monotonically reduced by increasing $T$?**: From Equation (4), we know that $g_{\lambda}$ is an increasing function. It is important to note that $g_{\lambda}$ is not simply composed with itself; as stated in line 166 of Theorem 2.3, it is composed in a specific manner. For example, after one composition, we have $g(2\lambda g_{\lambda}(\eta) + \eta)$, which is greater than $g_{\lambda}(\eta)$ because $2\lambda g_{\lambda}(\eta) + \eta > \eta$, and $g_{\lambda}$ is increasing. Repeating this process shows that the composition $[g(2\lambda(.) + \eta)]^{oT}$ always increases (or at least remains constant) with respect to $T$. Also, it is not intuitively plausible for the error to reduce as it propagates.
>
> - **In [WLZ22], Theorem 1 shows that the target error is bounded by the source risk and accumulated error w.r.t. domain number $T$, which is induced by gradual adaptation process. Note that the main result in submission show the accumulated error is a factor of source risk, which implies this result could be loose when the factor is large. Some comparison between the tightness of these bounds are highly appreciated.**: The reviewer is correct in noting this, and analyzing the tightness of these bounds and providing a lower bound for the error propagation would indeed be a valuable contribution. However, this analysis is beyond the scope of the current paper. We will address this in the future work section of the camera-ready version. Additionally, it is worth noting that in the opposite scenario, where the initial error is low, our bounds—even under linear error propagation—are still better than those in [WLZ22], as their rate of increase is independent of the initial error.
>
> - **Minor issues**: Thank you for pointing out these minor issues. All have been corrected in the revised version.
>
> ------------------
>
> We hope these revisions address the majority of the reviewer’s concerns and lead to a favorable reassessment. Once again, thank you for your timely and thorough review of our work.

---

> > ### Comment · Reviewer_C19w · 2024-08-11
> > **Concerns are addressed.**
> >
> > I thank the authors for their detailed responses. In the previous round review, my main concerns are the (theoretical) comparison with existing GDA theory and the tightness of bounds. In rebuttal, I found that 1) the advantages and weaknesses of derived bound over existing works are properly discussed; 2) the implications from derived results, e.g., Thm. 2.3 and Cor. 2.4, are further provided.
> >
> > Therefore, considering the modifications and improvements above, I'm satisfied with the responses and willing to improve the score.

---

> > > ### Author Response · Authors · 2024-08-11
> > > **Response to Reviewer C19w**
> > >
> > > Once again, we sincerely thank the reviewer for their time and effort in reviewing our paper. We are glad that the reviewer's concerns have been addressed and appreciate their decision to raise their score in favor of our submission.

---

### Official Review · Reviewer_BZqU · 2024-07-12

**Soundness:** 3
**Presentation:** 3
**Contribution:** 4
**Rating:** 7
**Confidence:** 4

**Summary:**

This paper proposes an optimization paradigm for gradual domain adaptation by iteratively performing manifold-constrained Wasserstein DRO and pseudo-labeling on the sequence of domains. The error propagation is theoretically investigated by a compatibility measure $g(\eta)$ between the manifold of distributions and the class of classifiers. It is shown that with sub-linear $g(\eta)$, the error propagation will be bounded by Wasserstein distance between adjacent domains, which is independent of the steps of adaptation. The theory is applied to analyze both gaussian generative models and a more general class of distributions characterized by an expansion property.

**Strengths:**

The paper makes good contribution in obtaining the first error bound for gradual domain adaptation that does not scale with $T$, blocking error propagation. The manifold assumption is natural and prevalent. And by the measure of compatibility, the author manages to associate the error propagation rate with complexity of the manifold structure. An exact threshold for error propagation is also obtained as a linear compatibility function. And the classic setting of mixture gaussian models for binary classification is also solved without error propagation.

**Weaknesses:**

1. At the end of proof of theorem 3.1, the author says that one can only upper bound $\min \\{  \mathbb E[\ell(x,y)], 1-\mathbb E[\ell(x,y)] \\}$, which is much a weaker result. Should the conclusion of theorem 3.1 be revised accordingly? Also can the minimum form be improved?
2. Theorem 3.1 is not tight in the current form for small distances between adjacent domains. Consider $\eta=0$ and $\lambda = 0$,
the error should always be that of Bayes optimal across domains, which is smaller than the current constant upper bound.
3. Theorem 3.2 is not tight for $\eta > 1$, in which case it is actually saying the upper bound of error rate will shrink.

**Questions:**

1. Typos: L126, reference missing. L186, citation missing. L228, $P_i$.
2. Is definition 4.1 correctly presented? The left hand side of the inequality is independent of the Borel set A, while the right hand side is dependent on A.

**Limitations:**

The author has not explicitly addressed the limitations of the work.

---

> ### Author Rebuttal · Authors · 2024-08-03
>
> We would like to thank the reviewer for their positive feedback. We have addressed the remaining concerns as follows:
>
> **Weaknesses**:
>
> - **At the end of proof of theorem 3.1, the author says that one can only upper bound $\min\\{\mathbb{E}[\ell(x, y)], 1- \mathbb{E}[\ell(x, y)]\\}$, which is much a weaker result. Should the conclusion of theorem 3.1 be revised accordingly? Also can the minimum form be improved?**: This limitation arises specifically in Theorem 3.1, where we consider the $(0-1)$-loss function. In this case, $1 - \mathbb{E}[\ell(x, y)]$ represents the expected loss of our classifier on $(x, -y)$. Essentially, this indicates that our classifier can separate the two classes very well, though it’s possible that the labels are flipped. Such guarantees are not uncommon in the literature, as discussed in [WSCM20]. Additionally, it should be noted that a few number of labeled data point from $P_T$ can always correct this label flipping.
>
> - **Theorem 3.1 is not tight in the current form for small distances between adjacent domains. Consider $\eta = 0, \lambda = 0$, the error should always be that of Bayes optimal across domains, which is smaller than the current constant upper bound.**: Theorem 3.1 was originally presented for cases where $\eta > 0$ and $\lambda \neq 0$. However, as the reviewer suggested, when $\eta = 0$, the error indeed corresponds to the exact Bayes error. To elaborate: as seen from the first line of inequalities in (37), when $\eta=0$, we have:
>
> $$
> g_{\lambda}^0(\eta)
> \leq \\, \inf_{\theta\in\Theta} \\, \sup_{P_{\mu}: \Vert \mu - \mu_0\Vert \leq 2\eta} E_{P_{\mu}} \\, [\ell(y,h_{\theta}(X))]
> = \inf_{\theta\in\Theta} E_{P_0} [\ell( y , h_{\theta} (\boldsymbol{X}))]
> = E_{\text{Bayes}},
> $$
>
> where $E_{\text{Bayes}}$ denotes the Bayes error. (We apologize for the cumbersome formulation, but this was necessary to work within the constraints of OpenReview). We will ensure that these edge cases are discussed in more detail in the camera-ready version.
>
> - **Theorem 3.2 is not tight for $\eta > 1$, in which case it is actually saying the upper bound of error rate will shrink.**: The reviewer is correct. As seen in the proof of Theorem 3.2, our analysis primarily focuses on cases where $\eta$ is not too large, which aligns with the typical scope of gradual domain adaptation research. When $\lambda$ is chosen moderately, $\eta \geq 1$ can significantly diminish the information of labels. For larger values of $\eta$, which were not the primary focus of this paper, alternative bounds and methodologies may be required to obtain more appropriate results.
>
>
> ---------------------
>
> **Questions**:
>
> - **Typos...**: We have carefully revised the paper to address this issue and assure the reviewer that all typos and rushed phrasing have been corrected.
>
> - **Is definition 4.1 correctly presented? The left hand side of the inequality is independent of the Borel set A, while the right hand side is dependent on A.**:  Thank you for highlighting this issue. Our intention was to show that for all Borel sets $A \in \mathcal{A}$, the two conditions hold. We will correct this by removing the $\inf$ and $\sup$, and explicitly stating that the conditions apply for $\forall A$.
>
>
> ------------------
>
> **Limitations: The author has not explicitly addressed the limitations of the work.**: We appreciate the reviewer’s input on this point. We will include a discussion of the limitations of our work, taking the reviewer's comments into careful consideration.
>
> -------------------
>
> We hope these revisions address the majority of the reviewer’s concerns. Once again, thank you for your timely and thorough review of our work.

---

> ### Comment · Reviewer_BZqU · 2024-08-07
>
> I acknowledge and thank the author for their response. Overall, I believe this paper significantly contributes to the existing theoretical analysis of gradual domain adaptation, being the first, as far as I know, to obtain a non-expansive error bound as the number of domains T increases. Despite some limitations in the practical validation of the algorithm, I maintain my support for its acceptance.
>
> I would appreciate it if the author could improve the clarity of the paper by revising the statement of Theorem 3.1 and Definition 4.1, discussing the limitations explicitly, as well as thoroughly checking for any other typos.

---

> > ### Author Response · Authors · 2024-08-09
> > **Response to Reviewer BZqU**
> >
> > Thank you for your time and favorable assessment of our work. We greatly appreciate your comments and feedback. Based on your and the other reviewers' suggestions, we conducted additional experiments to demonstrate the implementability and applicability of our method in real-world tasks, where it outperforms a number of rival methods. We also corrected all the typos, including those mentioned by the reviewers.

---

### Official Review · Reviewer_FbyN · 2024-07-13

**Soundness:** 4
**Presentation:** 4
**Contribution:** 4
**Rating:** 7
**Confidence:** 3

**Summary:**

The paper presents a new approach to gradual domain adaptation using distributionally robust optimization (DRO). This approach provides theoretical guarantees for model adaptation across successive datasets by bounding the Wasserstein distance between consecutive distributions and requiring that these distributions lie on a manifold. The theoretical analysis demonstrates that the proposed approach controls the error propagation and improves generalization across domains. Additionally, the analysis of two specific settings shows that the proposed approach eliminates the error propagation completely.

**Strengths:**

- The paper is well-written.
- The considered problem is timely and important.
- To the best of my knowledge, the proposed approach and the presented theoretical analysis are new.
- The presented theoretical results, especially those on error propagation, are significant and have potential practical implications.

**Weaknesses:**

- Although this is a theoretical paper, the experimental study is very limited and hidden in Appendix D.
- The relationship to existing work, both the classical theory of domain adaptation (e.g., [*]) and on gradual domain adaptation, could be clarified. Specifically, the only work mentioned on gradual DA is [KML20].
- The paper could benefit from thorough proofreading. For example, there are broken links and missing references (e.g., lines 126 and 186).

[*] Ben-David et al. A theory of learning from different domains. Machine Learning, 79(1):151–175, 2010.

**Questions:**

- Could the authors discuss possible extensions to non-binary settings and settings where the domain shift is applied to the joint distribution (Z) rather than the marginal (X)?

**Limitations:**

- The limitations appear only in the form of the assumptions required for the theoretical results to hold. Discussing the implications of these assumptions would enhance the paper.

---

> ### Author Rebuttal · Authors · 2024-08-03
>
> We would like to thank the reviewer for their positive feedback. We have addressed the remaining concerns as follows:
>
> **Weaknesses**:
>
> - **Although this is a theoretical paper, the experimental study is very limited**: We have expanded the experimental section significantly (please refer to the global rebuttal and the attached PDF). We tested our method on a real-world dataset used in the [KML20] paper and achieved superior results. These additional experiments and comparisons will be included in the camera-ready version.
>
> - **The relationship to classical theory of domain adaptation (e.g., Ben-David et al.) and on gradual domain adaptation, could be clarified. Specifically, the only work mentioned on gradual DA is [KML20].**: In addition to [KML20], we also discussed other relevant works, such as [WLZ22, HWLZ23, WSCM20], among others. To better position our work within the literature, we have expanded our discussion on these methods as well as the classical work of Ben-David et al. (as mentioned by the reviewer). This will be detailed further in the camera-ready version.
>
> - **The paper could benefit from thorough proofreading**: We agree that some phrases were rushed. We have carefully revised the paper to address this issue and assure the reviewer that all typos and rushed phrasing have been corrected.
>
> ------------------
>
> **Questions**:
>
> - **Could the authors discuss possible extensions to non-binary settings and settings where the domain shift is applied to the joint distribution (Z) rather than the marginal (X)?**: In our work, as shown in Equation (2), the domain shift is indeed applied to the joint distribution $\mathcal{Z}=\mathcal{X}\times\mathcal{Y}$, not just the marginal distributions, thus addressing the second concern raised by the reviewer. Regarding the extension to non-binary settings, generalizing our method (including Theorem 2.3 and Corollary 2.4) is straightforward. However, examples involving Gaussian distributions and manifold assumptions require further statistical analysis, which we consider a valuable direction for future work. Thank you for this suggestion, we will ensure that this topic is discussed in more detail in the camera-ready version.
>
>
> - **The limitations appear only in the form of the assumptions required for the theoretical results to hold. Discussing the implications of these assumptions would enhance the paper.**: The assumptions in our paper are, in essence, quite similar to those in [WSCM20] and [KML20]. These references have already discussed and experimentally validated the relevance of such assumptions in real-world datasets. We will elaborate on this issue further in the camera-ready version.
>
> ----------------
>
> We hope these revisions address the majority of the reviewer’s concerns. Once again, thank you for your timely and thorough review of our work.

---

> > ### Comment · Reviewer_FbyN · 2024-08-11
> >
> > I appreciate the authors' detailed responses and the additional experiments they conducted. I find them satisfactory and will therefore maintain my score.

---

> > > ### Author Response · Authors · 2024-08-11
> > > **Response to Reviewer FbyN**
> > >
> > > Once again, we sincerely thank the reviewer for their time, effort in reviewing our work and their recommendation for acceptance.

---

### Author Rebuttal · Authors · 2024-08-05

We would like to express our gratitude to all the reviewers for their thoughtful comments and feedback. As some questions and concerns were raised by multiple reviewers, we have provided a global response.

Some reviewers expressed concerns regarding the implementability of our method and its performance on real-world data, particularly under the manifold assumption. To address this, we have included a schematic in Figure 1 of the attached PDF, illustrating the workings of our method. As depicted, at the $i$th step, we perturb the data samples $(X_j,y_j),~j \in [n_i]$ from $P_i$ using a parametric function class, denoted as $f_P$, and penalize the extent of perturbation using the following term:

$$\frac{\gamma}{n_i}\sum_{j=1}^{n_i}{\Vert f_P(X_j) - X_j\Vert_2}.$$

These perturbed samples are then classified using a classifier. Let $L_C(f_P;X_1,\ldots,X_{n_i})$ represent the cross-entropy loss of the classifier on the perturbed samples. Our objective is to minimize $L_C(f_P;X_1,\ldots,X_{n_i}) -  \frac{\gamma}{n_i} \sum_{j=1}^{n_i}\Vert f_P(X_j) - X_j\Vert_2$ with respect to the parameters of the classifier $f_C$, while simultaneously maximizing it with respect to the parameters of $f_P$.

Our experimental details are as follows:

- We implemented this method on the 'Rotating MNIST' dataset, similar to [KML20]. In particular, we sampled 6 batches, each with a size of 4200, without replacement from the MNIST dataset, and labeled these batches as $D_0, D_1, \ldots, D_4$, which represent the datasets obtained from $P_0, P_1, \ldots, P_4$. The images in dataset $D_i$ were then rotated by $i \times 15$ degrees, with $D_0$ serving as the source dataset and $D_4$ as the target dataset. We provided the source dataset with labels and left $D_1, D_2, D_3$, and $D_4$ unlabeled for our algorithm. We then tested the accuracy of $\theta^*_0, \ldots, \theta^*_3$—the outputs of our algorithm at each step—on $D_1, D_2, D_3$, and $D_4$, respectively.

- We also implemented the GDA method exactly as described in [KML20]. For our method, we employed a 2-layer CNN with a $7\times 7$ kernel in the first layer and a $5\times 5$ kernel in the second layer. We also utilized an affine grid and grid sample function in PyTorch for $f_P$, following the approach introduced in [1]. For the classifier $f_C$, we used a 3-layer CNN with max pooling and a fully connected layer, applying dropout with a rate of 0.5 in the fully connected layer.

We compared our method to the GDA method presented in [KML20] and detailed the results in Figure 2 of the PDF. Additionally, we reported the accuracy of $\theta^*_0$ on $D_0$ as an example of in-domain accuracy. Our results show that our method outperforms GDA by a significant margin of 8 percent in the last domain.

[1] Jaderberg, M., Simonyan, K., & Zisserman, A. (2015). Spatial transformer networks. Advances in Neural Information Processing Systems, 28.

---

### Decision · Program_Chairs · 2024-09-25

**Decision:**

Accept (poster)

**Comment:**

The paper presents a new approach for gradual domain adaptation based on distributionally robust optimization. Some theoretical guarantees are provided with notably a bound showing that the error can be controlled and generalization improved.

In the original assessment, reviewers were positive but raised some issues on different aspects of the contributions.
During rebuttal, authors have addressed all the points mentioned by the reviewers.
In the discussion, all the reviewers provided a rather positive feedback on the paper.
A reviewer notably mentioned that the contribution provides the first establish a non-expansive error bound for an increasing number of domains.

Overall, all reviewers were positives, I recommend thus acceptance.

Nevertheless, I strongly recommend the authors to take into account all the feedbacks and include the additional elements provided in the rebuttal, as well as correcting the different typos.